# Review article: Towards Improved Drought Prediction in the Mediterranean Region – Modelling Approaches and Future Directions

Bouchra Zellou[1], Nabil EL Moçayd[2,3], EL Houcine Bergou[1]

[1] School of Computer Science, Mohammed VI Polytechnic University, Benguerir. 43150. Morocco

[2] International Water Research Institute, Mohammed VI Polytechnic University, Benguerir. 43150. Morocco

[3] Institute of Applied Physics, Mohammed VI Polytechnic University, Benguerir. 43150. Morocco

*Correspondence to:* Bouchra Zellou (bouchra.zellou@um6p.ma)

**Abstract.**

There is a scientific consensus that the Mediterranean Region (MedR) is warming and as the temperature continues to rise, droughts and heat waves are becoming more frequent, severe, and widespread. Given the detrimental effects of droughts, it is crucial to accelerate the development of forecasting and early warning systems to minimize their negative impact. This paper reviews the current state of drought modeling and prediction applied in the MedR, including statistical, dynamical, and hybrid statistical-dynamical models. By considering the multifaceted nature of droughts, the study encompasses meteorological, agricultural, and hydrological drought forms and spans a variety of forecast scales, from weekly to annual timelines. Our objective is to pinpoint the knowledge gaps in literature and to propose potential research trajectories to improve the prediction of droughts in this region. The review finds that while each method has its unique strengths and limitations, hybrid statistical-dynamical models appear to hold the most promising potential for skillful prediction with seasonal to annual lead times. However, the application of these methods is still challenging due to the lack of high-quality observational data and the limited computational resources. Finally, the paper concludes by discussing the importance of using a combination of sophisticated methods such as data assimilation techniques, machine learning models, and copula models and integrating data from different sources (e.g., remote sensing data, in-situ measurements, and reanalysis) to improve the accuracy and efficiency of drought forecasting.

**Key Words:** drought, forecasting, data assimilation, machine learning, Mediterranean, review

## 1 Introduction

Drought is a recurrent phenomenon in the Mediterranean Region (MedR). Throughout time, adaptation to this kind of climate event has been an important issue for the development of many countries in the region. Yet, with the disruptive accelerated impact of global warming, already reflected in more regular and intense droughts around the Mediterranean in the last few decades, building resilience to extreme weather conditions remains a true challenge (Satour et al., 2021). For these reasons among others, the region is often described as a hotspot for climate change (Tuel and Eltahir, 2020). The Intergovernmental Panel on Climate Change (IPCC) pointed out in the Sixth Assessment Report (AR6) that global warming has been more rapid in the Mediterranean than in the rest of the world (IPCC, 2021). This report projected an increase in the frequency and/or severity of agricultural and ecological droughts across the Mediterranean and Western Africa (IPCC, 2021). A global increase of 2 °C is thought to correspond to a 3 °C increase in the daily maximum temperature in the MedR (Seneviratne et al., 2016;

Vogel et al., 2021). If this increase in temperature continues at the same pace, MedR is susceptible to experience fearful desertification by the end of the 21$^{st}$ century, driving an increase in aridity (Carvalho et al., 2022).

This will surely lead to irreversible biodiversity loss and diminish the capability of semi-arid Mediterranean ecosystems to function as effective carbon sinks in the future (Valentini et al., 2000; Briassoulis, 2017; Zeng et al., 2021). These conditions exacerbate water stress, which, in turn, enhances the probability of wildfire (Turco et al., 2017a). A phenomenon already witnessed these two last summers (2021 and 2022) in several Mediterranean countries (Turkey, Greece, Italy, Algeria, and Morocco), displacing thousands, killing hundreds, and causing irreparable damage (Rodrigues et al., 2023; Yilmaz et al., 2023; Eberle and Higuera Roa, 2022).

The Mediterranean Sea (MEDS), lying between Africa, Europe, and Asia, serves as a substantial source of moisture and heat, affecting atmospheric circulation and weather patterns (Mariotti et al., 2008). Its narrow connection to the Atlantic Ocean via the 14 km wide Strait of Gibraltar and the surrounding varied topography (Fig. 1), with vegetated areas to the north and desert areas to the south and east, contribute to the region's complex climate dynamics (Michaelides et al., 2018).

The MedR is characterized by a mid-latitude temperate climate with mild rainy winters and hot, dry summers (Lionello et al., 2023). Notably, this area is positioned in a transitional band between the midlatitude and subtropical regions, which makes climate modeling for this region quite challenging (Planton et al., 2012). The Mediterranean climate exhibits a strong spatial gradient in precipitation, with generally decreasing precipitation values towards the south and hardly any precipitation during the summer (Lionello, 2012). Such conditions pose challenges in climate modeling and can lead to severe impacts on water supply and agriculture, especially in regions relying on rain-fed agriculture (Tramblay et al., 2020).

Water availability is unevenly distributed among the Mediterranean countries with 72% in temperate countries of the North, against 5% in the South, and 23% in the East (Milano et al., 2013). Accordingly, several countries such as Algeria, Morocco, Egypt, Libya, Malta, and some countries of southern Europe such as Portugal and Spain are experiencing a structural water shortage that is likely to increase with the expected population growth (Sanchis-Ibor et al., 2020). This situation is further aggravated when multi-annual droughts hit the region. In this challenging context, drought forecasting that provides seasonal to annual lead times becomes critically important for proactive agricultural and water resources management.

Growing concern about the drought phenomenon in the last decades has spurred the development of improved systems that predict the full cycle of drought (onset, duration, severity, and recovery) via a large number of indices and models. Common approaches to predicting drought can be subdivided into two categories of models: statistical models and dynamical models. Statistical models, also named data-driven models, rely on the estimated correlations between several predictors (large-scale climate variables) and predictands (local climate variables represented by historical observations). The climatology-based or persistence-based models, like the Ensemble Streamflow Prediction (ESP) system, form an essential tool in this category, leveraging both historical and near real-time data to generate a probabilistic forecast of future drought events (AghaKouchak, 2014a; Turco et al., 2017b; Torres-Vázquez et al., 2023). Meanwhile, dynamical drought prediction relies on the use of Global Climate Models (GCMs) to simulate the dynamical processes that govern hydroclimatic variability. Nevertheless, despite the usefulness of these models in drought prediction and early warning systems, their forecast accuracy remains

limited for longer lead times (exceeding one month) (Wood et al., 2015). The post-processing and multi-model ensemble techniques are usually used to improve prediction skills by avoiding systematic bias related to the coarse resolution of GCMs (Han and Singh, 2020). Recently, drought prediction has also been tackled by the hybrid statistical-dynamical models which combine the two approaches mentioned above. These models constitute a promising tool for long lead-time drought forecasting (Ribeiro and Pires, 2016).

Despite the efforts made to predict drought phenomena, it remains largely little understood due to its multiple causing mechanisms and contributing factors (Kiem et al., 2016; Hao et al., 2018). The complexity and variability depicted by many physical mechanisms such as Sea Surface Temperature (SST), North Atlantic Oscillation (NAO), El Niño—Southern Oscillation (ENSO), Mediterranean Oscillation (MO), and land-atmosphere feedback are also responsible for the low performance of drought monitoring and forecasting (Ayugi et al., 2022). Understanding the synoptic conditions leading to the drought phenomenon becomes increasingly important given the upward trend in temperature in the MedR. Further investigations to assimilate how large-scale teleconnections affect local weather and climate anomalies, as well as how these later feedback into the larger context, are much needed in this context.

To address these questions, numerous review papers have sought to consolidate the scientific advances in drought prediction from different regions of the world (e.g., Mishra and Singh, 2011; Hao et al., 2018; Fung et al., 2019; Han and Singh, 2020). While these studies provided a comprehensive overview of drought prediction at a global scale, our paper offers an in-depth analysis of drought prediction methodologies specifically applied to the Mediterranean context. This is achieved through an examination of the applicability, strengths, and limitations of statistical, dynamical, and hybrid statistical-dynamical models, in line with the regional specifics of the MedR. This specificity is vital given that drought, as a phenomenon, is highly region dependent. The unique meteorological conditions of the MedR necessitate dedicated studies, as solutions developed for other regions may not be applicable or effective here.

Tramblay et al. (2020) emphasized the urgent need for drought modeling and forecasting methods designed for the Mediterranean context, particularly as climate change continues to exacerbate drought conditions in this region. Building on this, our work not only emphasizes the complexities of drought assessment but also conducts a critical review of recent drought forecasting methodologies applied specifically to the MedR. In addition to shedding light on the merits and limitations of these methods, our investigation also helps identify underexplored areas that warrant further research. Detecting these gaps is a crucial aspect of our work, as it directs future research towards these relatively unexplored realms of drought prediction.

The structure of this paper is as follows: Section 2 highlights the difficulty related to the definition of drought from different perspectives. The causes of drought in MedR are provided in section 3. Sections 4, 5, and 6 present the recent advances in drought prediction with statistical, dynamical, and hybrid statistical-dynamical models respectively. Section 7 discusses the results found in this review, providing insights into the current state of drought forecasting in the MedR and highlighting potential areas for improvement. The challenges in drought prediction are reviewed with the prospects in section 8. Finally, the 9[th] section presents the conclusions of the whole paper.

**Figure 1 Topography of the Mediterranean Region** (30°N - 46°N in latitude and 10°W - 40°E in longitude)**.**

## 2    Drought Definitions, Classification, and Indices

Drought is a compound phenomenon of creeping nature. Establishing an accurate prediction, well describing its starting date and duration is extremely hard. The multidisciplinary and multiscale nature of drought renders the understanding of this phenomenon very challenging (AghaKouchak et al., 2021). As a matter of fact, literature gives numerous definitions for drought.

In the eighties, Wilhite and Glantz (1985) found more than 150 published definitions of drought that can be categorized into four broad groups: meteorological, agricultural, hydrological, and socioeconomical. This classification based on both physical and socioeconomic factors is still adopted today. As this classification is human-centered, some recent works emphasized the need to consider the ecological drought as well, which creates multiple stresses in natural ecosystems, see for example Crausbay et al. (2017), Vicente-Serrano et al. (2020), Bradford et al. (2020) and Zhang et al. (2022). Since the aim of this study is to review forecasting drought methods, we will focus only on the first three categories that provide direct methods to quantify drought as a physical phenomenon.

In an attempt to associate a mathematical definition with each drought type, several drought indices have emerged. These indices are typically based upon some hydroclimatic variables or parameters (indicators) such as temperature, precipitation, soil moisture, streamflow, and snowpack to describe three major characteristics of the drought event: severity, duration, and frequency. However, the lack of a universal definition of drought is also apparent in the huge variety of indices (more than 100) that have been developed for drought prediction (Lloyd-Hughes, 2014). Unfortunately, this plethora of indices creates more confusion than clarity (Lloyd-Hughes, 2014) and makes the choice of the most suitable indices a difficult task.

## 2.1. Meteorological Drought

The World Meteorological Organization (WMO) characterizes meteorological drought as "a prolonged absence or marked deficiency of precipitation". Similarly, the IPCC defines meteorological drought as "a period of abnormally dry weather in a region over an extended period". The threshold to distinguish between a dry or wet period often depends on the average rainfall typical for the specific area under study. This gives rise to a variety of meteorological definitions, each tailored to the distinct conditions of diverse regions or countries (Isendahl, 2006). Regarding the MedR, creating a single encompassing definition of meteorological drought is particularly challenging. This complexity stems from the diverse climate conditions across the region, particularly the pronounced variability between eastern and western meteorological conditions that contribute to drought.

The Standardized Precipitation Index (SPI) (McKee et al., 1993) and the Standardized Precipitation Evapotranspiration Index (SPEI) (Vicente-Serrano et al., 2010a) are two of the most prevalent indicators used to describe meteorological drought. They owe their popularity to the recommendation of the WMO (Svoboda et al., 2012). The SPI has been extensively used in previous studies for its ease of computation, its probabilistic nature, and its ability to detect drought at multiple time scales (Madadgar and Moradkhani, 2013; Chen et al., 2013; Li et al., 2020; Mesbahzadeh et al., 2020; Das et al., 2020). By fitting a probability distribution to observed precipitation data, the SPI is calculated and subsequently transformed into a standard normal distribution with a mean of 0 and a standard deviation of 1 (Livada and Assimakopoulos, 2007). Consequently, SPI values can be compared across various regions and timeframes (e.g., 1, 3, 6, 12, or 24 months). This multiscale nature of SPI enables it to capture diverse aspects of drought depending on the selected time scale. The shorter time scales (1-3 months) are suitable

for monitoring agricultural drought, while longer time scales (6-12 months or more) are better suited for evaluating
hydrological drought. However, it should be noted that the SPI considers only precipitation data and neglects the
variability of temperature and potential evapotranspiration (PET), ignoring the effect of warming on droughts.
Indeed, in relatively wet regions, precipitation deficit can constitute an important indicator for drought (Gamelin
et al., 2022). Yet, in midlatitude (or extratropic) regions such as the Mediterranean where the climatological
precipitation is modest or low, precipitation deficit may not be sufficient to measure extreme droughts.
Furthermore, knowing the upward trend in temperature and the influence of high atmospheric evaporative demand
(AED) in increasing severity of recent drought events in the MedR (Tramblay et al., 2020; Mathbout et al., 2021;
Bouabdelli et al., 2022), the choice of drought indices needs to prioritize those including these variables in their
formulation such as SPEI, or Palmer Drought Severity Index (PDSI) (Palmer, 1965) and Reconnaissance Drought
Index (RDI) (Tsakiris and Vangelis, 2005) to mention but a few.
The SPEI was developed by Vicente-Serrano et al. (2010a) using the climatic water balance concept of climatic
water supply and AED. It is based on precipitation and PET and has the advantage of combining the multi-scalar
character of the SPI with the ability to include the effects of temperature variability (Vicente-Serrano et al., 2010a).
A global assessment of drought indices conducted by Vicente-Serrano et al. (2012) found that SPEI provided a
superior capability in capturing drought impacts, particularly during the crucial summer season. Bouabdelli et al.
(2022) used SPI and SPEI indices and copula theory to study the impact of temperature on agricultural drought
characteristics under future climate scenarios over seven vast Algerian plains located in the MedR. The results of
this study confirmed that the frequency of drought events is much higher using SPI while their duration and severity
are more intense using SPEI. Russo et al. (2019) performed drought characterization in MedR using both SPEI and
SPI, considering the period 1980–2014. Their findings indicated that SPEI exhibits a stronger correlation with
drought conditions over a 3-month time scale, while SPI shows a better correlation for a 9-month duration. This
result highlights the ability of SPEI to capture the early shifts in the balance between evapotranspiration and
precipitation more efficiently than SPI (Russo et al., 2019).
Despite the utility of SPEI in drought characterization, it does have a noteworthy limitation. The effectiveness of
SPEI significantly relies on the method used for estimating PET such as the Penman-Monteith equation, the
Thornthwaite method, the Hargreaves method, and the Priestley-Taylor method among others. These estimation
methods can yield varying results, leading to inconsistencies in SPEI values. In essence, the sensitivity of SPEI to
the PET estimation method used could potentially affect the accuracy and reliability of the index in representing
drought conditions (Vicente-Serrano et al., 2010b; Stagge et al., 2014).
The PDSI has also been widely used to quantify the drought characteristics for a given location and time. It includes
precipitation, temperature, and soil moisture data to estimate water supply and demand and to reflect long-term
drought. But it has shown some inconsistencies when used at various locations (Wells et al., 2004). A self-
calibrating variant of this index (scPDSI) was proposed by Wells et al. (2004) to automatically calibrates the
behavior of the index by replacing empirical constants in its computation with dynamically estimated values to
account for the variability of precipitation and the climate characteristics between locations (Wells et al., 2004).
Ionita and Nagavciuc (2021) evaluated the drought characteristics at the European level over the period 1901–2019
using SPI, SPEI, and scPDSI. The results based on SPEI and scPDSI show that the increase in mean air temperature
and PET are making central Europe and the MedR dryer, whereas Northern Europe is getting wetter. While results

based on SPI using only precipitation data did not reveal this drought variability. This underscores the findings of Vicente-Serrano et al. (2012), who emphasized the benefits of using more integrative indices like SPEI in understanding and predicting drought variability more effectively.

The MedPDSI, which is an update of the PDSI formulation in terms of its soil water balance to consider real evapotranspiration (based on reanalysis data instead of PET) in the MedR, has allowed an earlier identification of longer and more severe droughts (Paulo et al., 2012). Paulo et al. (2012) compared SPI, SPEI, PDSI, and MedPDSI in detecting drought characteristics in Portugal for the period 1941 to 2006. They concluded that PDSI and MedPDSI are likely to identify better the supply-demand dynamics and that they may be of great interest for drought warning applications, aiming namely at agriculture (Paulo et al., 2012).

## 2.2. Agricultural Drought

Agriculture is very sensitive to climate variation especially extreme weather. Due to its dependency on water availability, this sector is strongly impacted by drought events. In the Mediterranean Basin, agricultural practices span both rain-fed and irrigated systems. Rain-fed agriculture is prevalent, particularly for crops such as wheat and barley, while crops like olives and citrus fruits, such as oranges, often utilize controlled irrigation systems to supplement natural precipitation (Rodrigo-Comino et al., 2021). Regardless of the system employed, if meteorological drought lasts for a prolonged period, it can lead to a reduction in soil moisture to such a level that it harmfully affects crop production, especially during the active plant growth season (Wilhite and Glantz, 1985; Mishra and Singh, 2010). At this stage the agricultural drought sets in.

Therefore, in addition to meteorological factors, the agricultural drought definition is also related to the retention capacity of soil in the crop growth season (Kuśmierek-Tomaszewska and Żarski, 2021) which depends on crop types, soil characteristics, and soil management. All these indicators can be employed to develop relevant agricultural drought indices. Among them, we cite Crop Moisture Index (CMI) (Palmer, 1968); Soil Moisture Deficit Index (SMDI); Evapotranspiration Deficit Index (ETDI) (Narasimhan and Srinivasan, 2005); Normalized Soil Moisture index (NSMI) (Dutra et al., 2008) and Empirical Standardized Soil Moisture Index (SSMI) (Carrão et al., 2016).

The formulation of these indices integrates soil moisture data, leveraging a variety of assessment techniques, each with unique advantages. These include in-situ soil moisture probes, cosmic-ray neutron probes, and physically driven models such as the ISBA land surface model (Tramblay et al., 2019). Each of these techniques has distinct advantages and is suitable for different application contexts (Miralles et al., 2010; Martens et al., 2017). However, when faced with the scarcity of observed soil moisture data, remote sensing comes to the forefront. It furnishes extensive and frequent measurements of soil moisture characteristics, effectively supplementing areas where observed data falls short. Yet, it is crucial to be aware of the limitations of these tools. Despite its indispensable role, remote sensing is constrained by factors such as coarse temporal and spatial resolution, limited penetration depth, and incompatible governing hydrologic principles (Mohanty et al., 2017; Gruber and Peng, 2022). As an alternative, hydrological models have been commonly used to simulate and calibrate this variable in the context of agricultural drought forecasts (Hao et al., 2018). Mimeau et al., (2021) used a modeling framework to estimate soil moisture sensitivity to changes in precipitation and temperature at 10 plots located in southern France. They

concluded that the current climate change scenarios may induce longer periods of depleted soil moisture content,
corresponding to agricultural drought conditions.
In general, when soil moisture in the root zone reaches a critical level, farmers resort to irrigation to save crops
(Kang et al., 2000). However, nowadays agriculture consumes approximately 85% of global fresh water for
irrigation (D'Odorico et al., 2019; Tatlhego et al., 2022), which is expected to increase in the years to come by
growing population, increasing food consumption, and rising temperatures that accelerate PET and promote
hydrological stress.

## 2.3. Hydrological Drought

Unlike agricultural drought which is mainly affected by the depletion of soil moisture after a dry period, a lack of
precipitation impacts many components of the hydrological system in a river basin or watershed (streams,
reservoirs, and lakes). These define water availability that can be used for commercial navigation, generation of
hydroelectric power, irrigation of farmlands, industry, and domestic activities for several months after the
deficiency in precipitation. Consequently, hydrological drought lags behind the occurrence of meteorological and
agricultural droughts. This lag time is a characteristic of the watershed, which is defined based on many physical
drivers such as evapotranspiration capacity, soil properties, vegetation types, snow accumulation/melt, local water
management such as dams' construction and control, water supply operation rules, and irrigation strategy (Van
Loon and Laaha, 2015).
A hydrological drought is generally proclaimed when the water levels in streamflow, reservoirs, lakes, aquifers,
and other water storage systems fall below a specific threshold. Therefore, the hydrological drought prediction
necessitates the analysis of climate variables such as precipitation and temperature and initial catchment conditions
(e.g., snow cover, and soil moisture) (Hao et al., 2018).
In the Mediterranean Basin, a common tendency for water levels to drop in shallow lakes and aquifers has
motivated many researchers to study the hydrological drought in this region: Greece (Myronidis et al., 2012);
Turkey (Akyuz et al., 2012); Tunisia (Hamdi et al., 2016); Lebanon (Al Sayah et al., 2021); Italy (Di Nunno et al.,
2021); Portugal (Mendes et al., 2022); Algeria (Bouabdelli et al., 2022); Syria (Mohammed et al., 2022). The most
common hydrological drought indices include Palmer Hydrologic Drought Index (PHDI) (Palmer, 1965), the
Streamflow drought index (SDI) (Nalbantis, 2008), and Standardized Runoff Index (SRI) (Shukla and Wood,
255 2008).

As part of the effort made by Palmer in the sixties, the PHDI has been developed by using the same two-layer soil
model as the PDSI, but it applies a stricter criterion for determining the ends of drought to account for long-term
drought events that reduce surface and groundwater supply. Vasiliades and Loukas (2009) tested the Palmer
indices in a Mediterranean basin (in Greece) they concluded that these indices were successful in the identification
of drought severity of historical events, but they were unable to identify drought duration.
The SRI is an index that uses the same computational principles as SPI but uses monthly mean streamflow rather
than precipitation only to account for the hydrologic process that determines seasonal lags in the influence of
climate on streamflow (Shukla and Wood, 2008). Shukla and Wood (2008) compared the SRI and the SPI results
during drought events in a snowmelt region. They concluded that the SRI can be used as a complement to the SPI
for depicting hydrologic aspects of drought.

The SDI is also a simple index that uses the cumulative monthly streamflow volumes for a given hydrological year to predict wet and dry periods and identify the severity of a hydrological drought (Nalbantis, 2008). Bouabdelli et al. (2020) conducted a comparison study of the SPI and SDI, focusing on their characteristics across three watersheds in northwestern Algeria. Their analysis revealed a substantial similarity between meteorological drought events (as represented by SPI-12) and hydrological drought events (as indicated by SDI-6). This correlation emphasizes the sensitive and responsive nature of these basins to dry conditions, further illustrated by the swift transition from meteorological to hydrological drought events in the studied basins (Bouabdelli et al., 2020).

The application of hydrological drought indices appears to be very valuable. However, the main challenge in applying these indices lies in the requirement for a long-term series of climatic data. According to the WMO, up to 30 years of continuous rainfall data may be necessary for accurate drought index calculations (WMO, 1994). This condition is not always fulfilled which makes the rainfall-runoff transformation a difficult task (De Luca et al., 2022). Modern hydrological models can offer a valuable counterpart to existing climate-based drought indices by simulating hydrologic variables such as land surface runoff (Shukla and Wood, 2008).

## 3    Overview of the physical mechanisms causing drought in the Mediterranean region

It is difficult to determine the physical mechanisms causing droughts in the Mediterranean basin since the region covers a complex landscape with high topographic and climatic heterogeneity, strong land-sea contrasts, and high anthropic pressure (De Luca et al., 2022).

Considering the various forms of drought, meteorological droughts, characterized by a deficit in precipitation, are commonly recognized as marking the onset of drought conditions. This initial stage is intrinsically linked to precipitation predictability, which is driven by large-scale atmospheric motions such as Walker circulations and Rossby waves, influenced by factors like SST anomalies, radiative forcing changes (both natural and anthropogenic), and land surface interactions (Hao et al., 2018; Wood et al., 2015). However, due to the inherently chaotic nature of atmospheric circulation, predictability, particularly for meteorological droughts, tends to diminish beyond a one-month lead time. It is crucial to note that the reliability of these predictions can differ when considering other drought types (such as agricultural or hydrological droughts) or altering the forecast scale, with seasonal forecasts often displaying more reliability months in advance, while daily forecasts may face limitations from around two weeks.

The discovery of teleconnections between SST anomalies and hydroclimatic phenomena constitutes a major advance in drought forecasting and early warning (Wood et al., 2015). Notably, it is widely established within the scientific community that certain ocean-atmospheric teleconnections, such as ENSO, can profoundly influence the onset of drought conditions in various regions worldwide, particularly in the tropics (Ropelewski and Halpert, 1987; Shabbar and Skinner, 2004; Hoell et al., 2014; Vicente-Serrano et al., 2017). For instance, during the peak phase of El Niño or La Niña in the tropical Pacific, a corresponding change in precipitation patterns can be observed several months later in North American winter climate (Livezey and Smith, 1999; Hoerling and Kumar, 2003). This delayed impact provides a crucial window for predicting potential drought conditions with a long lead time exceeding one month (Johnson and Xie, 2010). Moreover, this lagged correlation allows for proactive drought management strategies, with the ability to anticipate and prepare for drought conditions based on forecasted ENSO

conditions. Nevertheless, drought predictability is seasonally and spatially variable. Typically, the accuracy of seasonal drought prediction is superior in the tropics, while it still challenging in the extra-tropics (Doblas-Reyes et al., 2013).

In the MedR, the response of climate to ENSO is complex. It varies over time and depends on the maturity of the ENSO state, and the co-occurrence with NAO (Kim and Raible, 2021; Brönnimann et al., 2007; Mariotti et al., 2002). Although many authors have found a non-negligible correlation between ENSO and precipitation anomalies in the MedR, it remains insignificant compared to the tropics (Mariotti et al., 2002).

In contrast, the NAO is commonly identified as a prominent factor influencing Mediterranean climate variability during the winter season (Ulbrich and Christoph, 1999; Vicente-Serrano et al., 2011; Kahya, 2011; Santos et al., 2014; Cook et al., 2016). It is important to note, however, that while acknowledging the profound impact of the NAO on the climate dynamics of the MedR, its predictability, especially on seasonal scales, continues to be a considerable challenge in the field of climate science (Czaja and Frankignoul, 1999; Saunders and Qian, 2002; Scaife et al., 2014; Dunstone et al., 2016).

During the positive phase of the NAO, below-average precipitation rates are observed over large parts of the northern and western MedR. While in the negative phase of NAO, the climate is wetter and warmer (Lionello, 2012). Kim and Raible (2021) analyzed the dynamics of multi-year droughts over the western and central Mediterranean for the period of 850–2099. This analysis suggests Mediterranean droughts from 850-1849 CE were mainly driven by the internal variability of the climate system, including elements like barotropic high-pressure systems, positive NAO phases, and La Niña-like conditions. Conversely, external forcing such as volcanic eruptions were found to be associated with wetter Mediterranean conditions. In the period 1850-2099 CE, however, anthropogenic influences amplified land-atmosphere feedback, leading to persistent dry conditions in the Mediterranean (Kim and Raible, 2021).

Paz et al. (2003) analyzed monthly mean Sea Level Pressure anomalies (SLP) from the 1958–1997 record over the Mediterranean Basin. They identified a significant anomalous SLP oscillation between North Africa (NA) and West Asia (WA) and concluded that the regional trend of the NAWA index could explain increased drought processes in the eastern Mediterranean after the late '70s, in relation to northern hemispheric circulation.

The climate heterogeneity in the Mediterranean area may also be explained by the regional Mediterranean Oscillation (MO) characterized by the opposite precipitation patterns between the eastern and western regions (Dünkeloh and Jacobeit, 2003). More recently Redolat et al. (2019) proposed a new version of MO that uses areas instead of observatories or isolated points. The new index which is referred to as the Upper-Level Mediterranean Oscillation index (ULMOi) is based on the differences in geopotential height at 500 hPa to improve the predictability of seasonal anomalies in the Mediterranean climate (Redolat et al., 2019). According to this study, ULMOi has reported higher confidence than the MO index for rainfall predictability (Redolat et al., 2019). Other teleconnections influencing the climate of MedR can be found in the reviews done by Paz et al. (2003) and Lionello (2012). Recent works have also shed light on the impact of Madden Julian Oscillation (MJO) on water availability in the region, especially during heavy rainy episodes, see for example (Chaqdid et al., 2023)

In conclusion, several complex factors that influence the predictability of drought are not yet fully understood,
especially those related to climate change. Therefore, more research on the physical mechanisms causing drought
in the MedR is needed to improve the predictability of drought forecasts.
Expanding our grasp of the physical factors causing drought in MedR, we will now delve into drought forecasting
models. By leveraging insights from these mechanisms, scientists have developed numerous approaches and
techniques including statistical, dynamical, and hybrid statistical-dynamical models to boost the accuracy and
trustworthiness of drought predictions.
## 4    Statistical Drought Prediction Methods
Once the major sources of predictability are identified, the task of the statistical models is to uncover the spatial
and/or temporal relationship between a set of these potential predictors and the predictand. When a large number
of predictors are identified within the same region, dimension reduction techniques like Principal Component
Analysis (PCA) or Linear Discriminant Analysis (LDA) can improve model accuracy and efficiency by reducing
the number of dimensions while preserving essential information. On the other hand, feature selection methods
such as decision trees or Random Forests can help eliminate irrelevant predictors. These approaches can prevent
overfitting, leading to enhanced model performance and interpretability (Hao et al., 2018; Ribeiro and Pires, 2016).
The next sections will present the frequently used data-driven models and how they were employed to predict
different types of droughts at different spatiotemporal resolutions in the MedR.
### 4.1. Time Series models
During the last few decades, several methods have been developed to analyze the stochastic characteristics of
hydrologic time series (Morid et al., 2007; Rafiei-Sardooi et al., 2018; Band et al., 2022; Zarei and Mahmoudi,
2020). Moving average (MA), Autoregressive (AR), and Autoregressive Integrated Moving Average (ARIMA)
are all linear models that analyze past observations of the same variable to predict its future values. Normality and
stationarity of observations are two of the basic assumptions of these time-series models. Therefore, if some trends
or seasonality are detected in observations, they should be removed before the modeling to avoid any drift in the
concepts to be captured.
ARIMA is the most frequently used time-series model (Zhang et al., 2003). The popularity of this model is related
to its ability to search systematically for an adequate model at each step of the model building (identification,
parameter approximation, and diagnostic check). This method is based on the concept that nonstationary data could
be made stationary by "differencing" the series (Box et al., 2015). The approach involved considering a value Y
at time point t and adding/subtracting based on the Y values at previous time points and adding/subtracting error
terms from previous time points. The formula can be written as:

$$Y_t = c + \varphi_1 Y_{t-1} + \cdots + \varphi_p Y_{t-p} + \theta_1 e_{t-1} + \cdots + \theta_q e_{t-q} + e_t, \qquad (1)$$

where:
$Y_t$ is the value of the variable at time t; c is a constant term; p and q are the orders of AR and MA models,
respectively; $\varphi_i$ and $\theta_i$ are model parameters; $e_{t-1} \dots e_t$ are the error terms.

The AR component captures the impact of past values on the current value, the I component handles any non-stationarity in the data (i.e., changes in the mean or variance over time) by "differencing" the time series, and the MA component captures the impact of random shocks or errors in the data.

The ARIMA model is generally expressed with the three terms p, d, and q. The order of differencing in the I component is denoted by the value of (d) in the ARIMA(p,d,q) notation. It represents the number of times that the data must be "differenced" to produce a stationary signal. The lag order (p) represents the number of prior observations having a strong correlation with the current observation. While (q) is the size of the moving window and is identified by determining the number of lag errors that have a significant impact on the current observation.

The SARIMA is a more specific version of ARIMA that includes a seasonal component, which takes into account the repeating patterns that occur at regular intervals (e.g., daily, weekly, monthly) in the data. This makes it more appropriate for forecasting seasonal time series data.

Bouznad et al. (2021) conducted a comparative analysis of ARIMA and SARIMA models using precipitation, temperature, and evapotranspiration data to assess seasonal drought conditions in the Algerian highlands. These models were compared based on their ability to replicate and forecast the data series accurately. The SARIMA model emerged as the better choice as it exhibited significant p-values for all variables under study. This implies that the model was statistically significant in predicting the variables and thus outperformed the ARIMA model in this specific context. In the same country, Achite et al. (2022) investigated the meteorological and hydrological drought in the Wadi Ouahrane Basin using ARIMA and SARIMA models applied to SPI and SRI indices. A validation based on R² revealed high accuracy for SPI and SRI of 0.96 and 0.97 respectively, at 1-month lag. Additional examples of the use of the time-series model in drought forecasting in MedR can be found in Table 1.

Although time series models have shown good predictability of drought characteristics, these methods present certain limitations as they are based solely on the persistence of some drought indicators (trend, seasonality) without worrying about their interactions.

**Table 1 Main studies using the Time series model to forecast drought in the MedR.**

### 4.2. Regression analysis

Regression models are commonly applied in drought forecasting due to their straightforwardness, interpretability, and proficiency in revealing potential connections between hydroclimatic variables. These models use various predictors (independent variables), including precipitation, temperature, and other relevant climate indices, to approximate drought indices or related target variables (dependent variables).

Simple and multivariate linear regression (MLR) models have been broadly applied for projecting extreme hydrological phenomena such as droughts (Sharma et al., 2018). These models shed light on the linear connections between various predictors and predictands, offering a valuable method to understand the primary factors of drought conditions and their interactions (Mishra et al., 2011).

An MLR model that predicts drought from multiple drought predictors $X_1, X_2, \dots X_n$ can be formulated as:

$$Y = \beta_0 + \beta_1 X_1 + \beta_2 X_2 + \cdots + \beta_n X_n + \varepsilon \qquad (2)$$

Where:

$\beta_0$ is the y-intercept or the constant term,

$\beta_{i(i=1,2,...,n)}$ are the regression coefficient for each independent variable $X_{i(i=1,2,...,n)}$,

$\varepsilon$ is the model's error term.

On the other hand, when drought forecasts have a binary or dichotomous nature, such as drought vs. no drought, logistic regression models can be particularly useful. In these cases, the dependent variable (drought) is expressed as a probability or likelihood of occurrence. The main goal of logistic regression is to estimate the relationship between a set of predictors and the probability of the binary outcome (Rahali et al., 2021; Hosmer et al., 2013).

Some of the applications of regression analysis for drought forecasting in the MedR are discussed below and summarized in (Table 2).

**Table 2 Main studies using regression analysis to forecast drought in the MedR.**

Sousa et al. (2011) analyzed the spatiotemporal evolution of drought conditions across the MedR during the 20[th] century using monthly precipitations, NAO, and SST as independent variables and scPDSI as a dependent variable. Their study successfully developed a robust stepwise regression model capable of predicting summer drought conditions six months in advance with a high correlation of 0.79 between simulated and observed scPDSI time series, thus demonstrating its utility in forecasting future drought conditions in the region. Tigkas and Tsakiris (2015) used the MLR model with variables that include the minimum temperature and RDI as the main independent variable for the assessment of drought effects on wheat yield in two rural areas of Greece. The results of this analysis showed a high correlation between RDI and the wheat yield during the winter months which proves that satisfactory prediction of the drought impacts on wheat yields 2 to 3 months before the harvest can be achieved using the MLR model. Martínez-Fernández et al. (2016) conducted a study in the REMEDHUS (Soil Moisture Measurement Stations Network) area in Spain, aiming to monitor agricultural drought on a weekly time scale and provide early warning to farmers for adapting irrigation strategies. They computed a specific agricultural drought index (SWDI) using data from the SMOS satellite. Within this study, various computation approaches were analyzed, and the ones that yielded the most promising results were those directly based on soil attributes or parameters extracted from pedo-transfer function (PTF). These approaches utilized a multiple regression analysis, with soil water parameters as dependent variables and incorporated other relevant soil characteristics such as texture, bulk density, and porosity.

Although regression models have been valuable in drought forecasting, they exhibit certain limitations such as the linearity assumption, limited interactions between variables, sensitivity to overfitting and multicollinearity (Rafiei-Sardooi et al., 2018). Consequently, their ability to accurately represent complex real-world phenomena is often insufficient (Zhang, 2003). To address these shortcomings, more advanced models capable of capturing non-linear relationships and interactions are required, ultimately improving the forecasting of complex hydroclimatic events such as droughts.

## 4.3. Machine Learning and Hybrid Models

One of the big challenges in drought prediction is the random and nonlinear nature of the hydroclimatic variables (Agana and Homaifar, 2017). Over the last two decades, intelligent techniques such as Artificial Neural networks (ANN), Support Vector Machines (SVM), and Fuzzy Logic (FL) have proven to be very promising tools for modeling nonlinear and dynamic time series (Mokhtarzad et al., 2017; Dikshit et al., 2022; Prodhan et al., 2022). These algorithms have thus garnered significant interest in the realms of drought modeling and forecasting (Prodhan et al., 2022). In the context of modeling, they are used to develop mathematical representations of complex drought systems, capturing the interplay of various atmospheric, hydrological, and land surface processes that lead to these phenomena. In forecasting, the models derived from these algorithms are employed to anticipate future drought conditions, assisting in risk assessment and mitigation strategies. Table 3 highlights key studies that utilize machine learning models for drought prediction in the MedR.

Prodhan et al. (2022) stated in their review of machine learning methods for drought hazard monitoring and forecasting on a global scale that the ANN was the most popular model in peer-reviewed literature, and they suggested that higher use of the ANN model is anticipated because it has non-linear properties that make it more robust for identifying all possible interactions between predictors.

ANN is a mathematical model inspired by biological brain neural networks. It consists of an interconnected group of nodes (artificial neurons) and processes information using a connectionist computation (Fig. 2). In the case of drought forecasting, ANN architecture is usually made of three layers: an input layer which consists of the drought predictors, hidden layer(s) which comprises a function that applies weights to the input variables and passes them using a non-linear activation function, and an output layer that consists of the drought target variable or drought index (Han and Singh, 2020).

**Figure 2 Drought forecasting based on a simple ANN architecture.**

For the proper functioning of a neural network, the optimization of network weights (known as the learning or training process) is an essential step (Dikshit et al., 2022). Back-propagation, Feed Forward, Gradient Descent, Stochastic Gradient Descent, Adam and Levenberg–Marquardt are among the common training algorithms (Bergou et al., 2020). The role of these algorithms is to minimize the difference between predicted and observed values by adjusting the network weights and biases of the model.

Di Nunno et al. (2021) used a non-linear AutoRegressive with eXogenous inputs (NARX) neural network (a particular type of recurrent dynamic ANNs) to predict spring flows in the Umbria region (Italy). The results of this study show a good performance of the NARX model in predicting spring discharges for both short (1 month: $R2 = 0.90$–$0.98$, $RAE = 0.09$–$0.25$) and long-term lag time (12 months: $R2 = 0.90$–$0.98$, $RAE = 0.09$–$0.24$). Achour et al. (2020) also confirmed the performance of the ANN model with multi-layer perceptron networks architecture and Levenberg–Marquardt calibration algorithm in predicting drought in seven plains located in northwestern Algeria with 2 months lead time ($R^2 = 0.81$, $RMSE < 0.41$ and $MAE < 0.23$).

SVM is also a robust supervised learning model that investigates data for classification and regression analysis. It designates the best separating line to classify the data with more safety margins. Besides, the good performance in solving linear problems, SVM could also transfer a non-linear classification to a linear one using the kernel function and be able to solve high-dimensional problems (El Aissaoui et al., 2021).

In the context of drought studies, SVM is particularly beneficial due to its ability to handle many inputs, use a small dataset for training, and its resistance to overfitting compared to ANN (Hao et al., 2018). These features make SVM less sensitive to data sample size, enhancing the robustness of the drought model. On the forecasting aspect, SVM employs a kernel function to map predictors in a high-dimensional hidden space, subsequently transforming the predictand to the output space (El Aissaoui et al., 2021). This process allows the SVM model to generate effective and accurate forecasts about potential future drought events, given the input variables.

El Aissaoui et al. (2021) used the Support Vector Regression (SVR) model with three kernel functions (linear, sigmoid, polynomial, and radial basis function [RBF]) for the prediction of drought in the region of Upper Moulouya (Morocco) through the SPI and SPEI indices. Their research underscores the SVR model's effectiveness, particularly with the RBF kernel function, in forecasting drought indices SPI (R = 0.92) and SPEI (R = 0.89). Mohammed et al. (2022) evaluated the applicability of 4 Machine Learning algorithms namely bagging (BG), random subspace (RSS), random tree (RT), and random forest (RF) in predicting agricultural and hydrological drought events in the eastern MedR based on SPI. The results of this study revealed that hydrological drought (SPI-12, −24) was more severe over the study area and BG was the best model in the validation stage with RMSE ≈ 0.62–0.83 and r ≈ 0.58–0.79.

To further improve the prediction accuracy of AI models, preprocessing of data using wavelet decomposition (WD), PCA, or empirical mode decomposition (EMD) is recommended. These techniques known as hybrid models have gained attention due to their potential to improve prediction accuracy and better capture complex relationships in the data (Yoo et al., 2015; Liu et al., 2020). The preprocessing techniques are used to extract and represent the essential features and patterns within the data and statistical methods, such as ANN, SVM, or RF, model the relationship between the input variables and the target drought index. El Ibrahimi and Baali (2018) explored the prediction of short-term (SPI-3) and long-term (SPI-12) drought conditions using 6 models: SVR, ANN-MLP, Adaptive Neuro-Fuzzy Inference Systems (ANFIS), WA-SVR, WA-MLP, and WAANFIS in the Saïss Plain (Morocco). They argued that ANN models were more efficient than SVR models and that the use of wavelet analysis has enhanced the prediction skill of ANN models which is probably due to their capacity in detecting local discontinuities and non-stationary characteristics of the data.

**Table 3 Main studies using Artificial Intelligence Models to forecast drought in the MedR.**

Özger et al. (2020) evaluated the effect of using EMD and WD for decomposing time series data on drought prediction using the self-calibrated Palmer Drought Severity Index (sc-PDSI) and machine learning models ANN and SVM. They found that the accuracy of standalone machine learning models in midterm sc-PDSI predictions was unsatisfactory, but it significantly improved when EMD and WD techniques were introduced, particularly for hybrid wavelet models.

In summary, machine learning and hybrid models, which combine preprocessing techniques with statistical methods, have demonstrated their efficiency in drought forecasting, as they can effectively handle intricate, nonlinear relationships and adjust to a diverse range of input data characteristics. However, the applicability of these models may be challenging when input variables exhibit strong dependence on each other. This dependency can lead to several issues such as multicollinearity, overfitting, and diminishing returns (Maloney et al., 2012). To

address these limitations and improve drought forecasting performance, it is essential to consider joint probability
models (Madadgar et al., 2014; Hao et al., 2018).

### 4.4. Joint Probability Models

The probabilistic analysis of drought events plays a significant role in the planning and management of water
resource systems, particularly in arid or semi-arid Mediterranean regions known for low annual and seasonal
precipitation. Drought return periods, which estimate the frequency of drought events, can provide valuable
information for responsible water management during drought conditions. The univariate frequency analysis is a
common method for analyzing drought events. As mentioned above, drought is usually characterized by its
severity, duration, and frequency which can be extracted using the theory of runs introduced by Yevjevich (1967).
These characteristics present a dependence structure that can be ignored by the univariate approach, resulting in
an under/overestimation of drought risks. As such, several joint probability theories have been recently
incorporated into drought risk analysis including two or more variables. One of the most important joint probability
models that have garnered increasing attention in the hydrologic community over the last decade is the copula
model (Jehanzaib et al., 2021; Pontes Filho et al., 2020; Das et al., 2020; Zellou and Rahali, 2019; Mortuza et al.,
2019; Ozga-Zielinski et al., 2016; Xu et al., 2015; AghaKouchak, 2014b; Madadgar and Moradkhani, 2013; Chen
et al., 2013).
There are numerous copula families and classes, such as elliptic, Archimedean (Clayton, Frank, Gumbel, Joe),
extreme value, and Bayesian to cite but a few. The choice of the most suitable copula family depends on the
specific modeling goals and the structure of the data being modeled (Genest and Favre, 2007 ; Joe, 2014).
A brief overview of the bivariate copula theory is given here to initiate readers about their concept and application.
However, for additional details on the theory and concepts of the copula, readers may refer to the monographs by
Joe (1997) and Nelsen (2007). Furthermore, comprehensive methodological understanding of constructing high-
dimensional copulas, such as Pair Copula Construction (PCC) and Nested Archimedean Construction (NAC), can
be garnered from the works of Aas and Berg (2009) and Savu and Trede (2010).
Let F be a 2-dimensional distribution function, with univariate margins $F_1$ and $F_2$ for random variables U and V,
respectively. According to Sklar's theorem (Sklar, 1959), there exists a copula C such that:

$$F(U, V) = C\big(F_1(U), F_2(V)\big) \; U, V \in R \qquad\qquad (3)$$

with $C$ unique when $F_1(U)$ and $F_2(V)$ are continuous marginal distributions, so that
$C: [0,1]^2 \rightarrow [0,1]$ that satisfies the boundary conditions $C(u, 0) = C(0, v) = 0$
and $C(u, 1) = C(1, u) = u$ (Uniform margins) for any $u \in [0,1]$ and the so-called 2-increasing property
(Papaioannou et al., 2016).
The main advantage of the copula over the traditional multivariate distributions is its ability to model the nonlinear
dependence structure between variables independently from the choice of their marginal distributions (Salvadori
and De Michele, 2004). This concept simplifies the joint probability analysis and its application in high dimensions
(with a large number of variables or predictors) becomes possible.

Serinaldi et al. (2009) constructed a four-dimensional joint distribution using the copula approach and SPI to model the stochastic structure of drought variables in Sicily (Italy). Drought return periods were next computed as mean interarrival time, taking into account two drought characteristics at a time by means of the corresponding bivariate marginals of the fitted four-dimensional distribution. Bouabdelli et al. (2020) investigated the joint probability and joint return period of drought severity and duration using copula theory to assess the hydrological drought risk in the reference period and its probability of occurrence in the future under two climate change scenarios in three basins located in northern Algeria. Bonaccorso et al. (2015) evaluated the conditional probability of future SPI classes under the hypothesis of multivariate normal distribution of NAO and SPI series in Sicily (Italy). The results of this study indicated that transition probabilities toward equal or worse drought conditions increase as NAO tends toward extremely positive values. Table 4 displays additional examples of the application of the Joint Probability Models to forecast drought in the MedR.

**Table 4 Main studies using Joint Probability Models to forecast drought in the MedR.**

All the above-mentioned studies confirm that copulas can accurately capture the joint distribution and dependence structure between multiple drought predictors without making strong assumptions about their marginal distributions. By combining the strengths of machine learning models with the flexibility of copulas, researchers can develop more accurate and reliable hybrid methods that better represent the intricacies of hydrological processes and climatic variables, even in the presence of strong dependence among the input variables (Jiang et al., 2023; Li et al., 2022; Wu et al., 2022; Zhu et al., 2020).

### 4.5. Ensemble Streamflow Prediction

The ESP method, a commonly used technique in hydrological forecasting, was primarily intended for medium to long-term streamflow prediction (Day, 1985). However, its utility extends to the prediction of hydrological droughts, characterized by low streamflows (KyungHwan and DegHyo, 2015 ; Sutanto et al., 2020 ; Troin et al., 2021).

ESP operates on the principle of employing historical data to generate an ensemble of possible future climate conditions (Turco et al., 2017b). The process begins by determining the current state of the system, considering parameters such as current streamflow, soil moisture levels, and reservoir levels which serves as the initial conditions for the forecast (Wood et al., 2016). The generation of the ensemble involves choosing a historical record at each time (day, week or month) of forecast that will provide the meteorological inputs (Day, 1985). By repeating this process for every time in the historical record, an ensemble of forecasts is produced, each member representing a potential future scenario. The hydrological model is run for each ensemble member, using the chosen meteorological inputs and initial conditions to generate a range of potential future states of the system (Harrigan et al., 2018). The ensemble of forecasts is then analyzed to derive probabilistic predictions.

As new data becomes available, forecasts can be updated by re-initializing the system's state and generating a new ensemble of forecasts. A significant advantage of this method is that it enables the uncertainty prediction by producing a variety of potential future streamflow forecast scenarios which can increase the confidence of this approach, specifically for its operational use in water management (Troin et al., 2021).

However, the limitations of the ESP method must be noted. For instance, it presupposes that future behavior will
mirror past behavior, a concept that may not hold under changing climatic conditions (Wood et al., 2016).
Furthermore, the method's performance is heavily reliant on the quality and duration of the historical
meteorological records used in the ensemble generation process (Turco et al., 2017b).
ESP is frequently employed as a benchmark for comparison with more sophisticated forecasting methods, such as
dynamical climate models or hybrid statistical-dynamical models (AghaKouchak, 2014a; Turco et al., 2017b;
Torres-Vázquez et al., 2023). Although these more complex methods can outperform ESP in some instances, the
computationally efficient ESP method often exhibits comparable performance, particularly when forecasting a few
months ahead (Turco et al., 2017b; Torres-Vázquez et al., 2023).

### 4.6. Markov Chain Models

Markov chains are effective tools to understand the stochastic characteristics of drought events and their temporal
dependency. These models assume that future states depend only on the current state.
Mathematically, Markov chain is a stochastic process $X$, such as at any time t, $X_{t+1}$ is conditionally independent
from $X_0, X_1, X_2, ..., X_{t-1}$, given $X_t$; the probability that $X_{t+1}$ takes a particular value j depends on the past only
through its most recent value $X_t$:

$$P\{X_{t+1} = j | X_0, X_1, ....., X_t\} = P\{X_{t+1} = j | X_t = i\} \forall i, j \in S, t \in T \qquad (4)$$

A Markov chain is characterized by a set of states, $S$, and by the transition probability, $P_{ij}$, between states. The
transition probability $P_{ij}$ is the probability that the Markov chain is at the next time point in state $j$, given that it is
at the present time point in state $i$.
The drought prediction using this concept can be expressed as the transition from wet or normal state to dry state
(or the other way around) or the transition from one drought severity state to another (e.g., no drought, mild
drought, moderate drought, extreme drought). Habibi et al. (2018) studied meteorological drought in North
Algeria's Chéliff–Zahrez basin, employing both localized and spatially distributed probabilities for temporal
transitions using Markov Chains, and recurrence probabilities using an optimal time series model, the APARCH
approach. Paulo and Pereira (2007) used Markov chains, incorporating homogeneous and non-homogeneous
formulations, to predict drought transitions up to three months ahead, based on the SPI derived from 67 years of
data in Southern Portugal. The non-homogeneous Markov model outperformed its counterpart by considering the
initial month and seasonal rainfall variations. Table 5 lists additional studies that apply Markov chain models for
MedR drought forecasting.
**Table 5 Main studies using Markov Chains Model to forecast drought in the MedR.**
These studies generally support the effectiveness of Markov chain models in providing valuable drought insights.
However, it is essential to consider the challenges associated with applying Markov chains within the MedR, as
the region's complex topography, considerable interannual climate fluctuations, limited data availability, and the
non-stationarity resulting from climate change can adversely affect the models' core assumptions and constrain
their long-term forecasting accuracy. Addressing these challenges calls for the adoption of more sophisticated
techniques that encompass both stochastic and physically based approaches, ultimately enhancing the accuracy
and reliability of drought predictions in this region (Paulo and Pereira, 2007).
**5    Dynamical Drought Prediction Methods**
Dynamical drought prediction methods are generally based on the use of seasonal climate forecasts derived from
comprehensive GCMs. The European Centre for Medium-Range Weather Forecasts (ECMWF)'s System 4
(SYS4), the Hadley Centre's Global Environmental Model (HadGEM), the Community Earth System Model
(CESM), and the National Centers for Environmental Prediction (NCEP)'s Climate Forecast System (CFS) are
some widely recognized examples. Designed to emulate physical processes across the atmosphere, ocean, and land
surface, these GCMs can produce near-term forecasts for various climatic factors such as precipitation,
temperature, surface pressure, and winds. However, these models typically provide a global overview and possess
a relatively coarse resolution, which spans from 150 km to 300 km horizontally, encompassing 10 to 20 vertical
atmospheric layers and up to 30 oceanic layers. This level of detail may not offer the specificity necessary for
local-scale impact assessments. To counter this, post-processing steps, encompassing downscaling and bias
correction, are crucial when employing GCM forecasts (Tuel et al., 2021; Gumus et al., 2023). The main objective
here is to refine the global, coarse-grained GCM data into higher-resolution forecasts. These refined forecasts are
far more pertinent for predicting seasonal drought events at a regional and local scale within the MedR.
The most common approaches to downscale GCM forecasts include statistical models, dynamic or nested models,
and hybrid statistical–dynamical models (Wilby et al., 2004). In statistical downscaling, large-scale variables are
used as the predictors and desired near-surface climate variables are the predictands (Gutiérrez et al., 2019). The
role of statistical models is then to measure the correlations between predictors and predictands. Whereas
dynamical downscaling refers to the use of high-resolution regional simulations to dynamically extrapolate the
effects of large-scale climate processes to regional or local scales based on a nesting approach between GCMs and
Regional Climate Models (RCMs) (Giorgi and Gutowski, 2015). However, it is known that GCMs contain
significant systematic biases that may propagate into RCMs through the lateral and lower boundary conditions
and thus degrade the dynamically downscaled simulations and lead to large uncertainties (Maraun, 2016). Besides,
climate predictions from a single climate model simulation are sensitive to initial oceanic and atmospheric states
and can represent only one of the possible pathways the climate system might follow.
**5.1. Multi-Model Ensemble**
To allow probabilistic estimates of climate variables with uncertainties in quantification, it is necessary to carry
out an ensemble of simulations with different initial conditions from each model and to combine various models
as ensemble members. The frequently used Multi-Model Ensemble (MME) and bias correction methods include
quantile mapping (Wood et al., 2002) and Bayesian Model Averaging (Krishnamurti et al., 1999; Seifi et al., 2022).
These methods proceed by adjusting the modeled mean, variance, and/or higher moments of the distribution of
climate variables, to match the observations. However, such MME simulations can be very computationally
demanding. Therefore, some international dynamical downscaling intercomparison projects were carried out such
as the Coordinated Regional Downscaling Experiment (CORDEX, Wilby et al., 1998) and its Mediterranean
initiative MEdCORDEX (Ruti et al., 2016) to provide present and future climate simulations with a high spatial
resolution (~12 km). In a study conducted by Turco et al. (2017b), the accuracy and reliability of ECMWF's System
4 (SYS4) in forecasting drought conditions, characterized by a six-month SPEI6, across Europe from 1981 to 2010
was evaluated. They found that the SYS4 model effectively projected the spatial patterns of SPEI6 and various
drought conditions (ranging from extreme to normal) with a reasonable degree of precision up to a lead time of 2
months. In the same geographical context, Ceglar et al. (2017) demonstrate the power of dynamical models in the
agricultural sector by investigating the relationship between large-scale atmospheric circulation and crop yields in
Europe. Their research highlights the significant potential of such models in developing effective seasonal crop
yield forecasting, and consequently, in advancing dynamic adaptation strategies to climate variability and change.
All these studies confirmed the good performance of MME methods in providing probabilistic drought forecasts
for 1 to 2 months of lead time and improving drought onset detectability. However, much effort should be made
in selecting the most skilled GCM ensembles in reproducing the large and synoptic scale atmospheric and land-
surface conditions associated with drought development in the MedR. By prioritizing ensembles that adequately
capture the region's distinct climate characteristics, spatial-temporal variability, and land-atmosphere interactions,
the MME forecasts can mitigate biases related to key meteorological variables such as temperature or precipitation
and significantly improve the precision and reliability of drought predictions (Li et al., 2023; Ahmed et al., 2019).
**5.2. Coupled hydrological models.**
On the other hand, GCMs often struggle to accurately represent some complex elements of the hydrological cycle,
such as soil moisture, streamflow, groundwater level, and PET. The inherent complexities of these variables and
the broad spatial scale of GCMs make it challenging to fully capture their behavior. This gap can limit the
effectiveness of GCMs in drought prediction and modelling (Balting et al., 2021). Consequently, to dynamically
forecast agricultural and hydrological droughts, the water balance should be correctly simulated by hydrological
models forced by climate forecasts (Wanders and Wood, 2016). Among the most used models to forecast
hydrological drought, we cite, the Soil and Water Assessment Tool (SWAT) (Arnold et al., 1998), the Variable
Infiltration Capacity (VIC) (Liang et al., 1994), and the Community Land Model (CLM) (Oleson et al., 2004).
These models can incorporate data on soil moisture, vegetation, snow water equivalent, groundwater level, and
other initial hydrologic conditions with climate forecasts to simulate the movement of water through the
hydrological cycle, including the processes of precipitation, evaporation, infiltration, and runoff. Crop growth
models can also be coupled with hydrological models to make an accurate prediction of agricultural drought and
its impact on crop yields (Narasimhan and Srinivasan, 2005; Abhishek et al., 2021).
Coupled hydroclimatic models can improve drought forecasting by allowing for the consideration of feedback
between the hydrological and climatological components of the Earth system. Indeed, drought conditions can
affect the availability of water for evapotranspiration, which in turn can affect the amount of moisture in the
atmosphere and the likelihood of precipitation. By incorporating this feedback into the model, it is possible to
produce more accurate forecasts of drought conditions.
In a recent study, Brouziyne et al. (2020) combined meteorological and hydrological drought indices (SPI and
SDI) with a SWAT model forced by bias-corrected CNRM-CM5 data to predict future droughts under two RCPs
(4.5 & 8.5) in Bouregreg watershed, Morocco. They confirmed that using multiple drought indices and a
comprehensive water budget indicator such as Total Water Yield provided a valid approach to evaluate drought

conditions in a Mediterranean context. Marx et al. (2018) analyzed a multi-model ensemble of 45 hydrological simulations based on three RCPs (2.6, 6.0, and 8.5), five GCMs (CMIP5), and three state-of-the-art hydrological models (mHM, Noah-MP, and PCR-GLOBWB) to investigate how hydrological low flows are affected under different levels of future global warming. Based on the analysis of the results, the authors recommended using multiple hydrological models in climate impact studies and to embrace uncertain information on the multi-model ensemble as well as its single members in the adaptation process.

## 5.3. Long-term drought projection under climate change.

As climate change continues to influence drought events in the MedR, it is vital to integrate long-term climate projections into drought forecasting strategies (Tramblay et al., 2020). In this regard, GCMs are essential for projecting future climate changes under varying scenarios, such as Representative Concentration Pathways (RCPs) or Shared Socioeconomic Pathways (SSPs[1]). Coupled with downscaling techniques, these models offer region-specific projections of critical climate variables including precipitation, temperature, surface pressure, and winds. These projections are instrumental in estimating long-term drought events, facilitating a more comprehensive risk assessment for stakeholders and decision makers. Baronetti et al. (2022) analyzed the expected characteristics of drought episodes in the near (2021–2050) and far (2071–2100) future compared to the baseline conditions (1971–2000) for northern Italy using EURO-CORDEX and MedCORDEX GCMs/RCMs pairs at a spatial resolution of 0.11 degrees for the RCPs (4.5 and 8.5) scenarios. The results indicated that the GCM/RCM pairs performed generally well, while in complex environments such as coastal areas and mountain regions, the simulations were affected by considerable uncertainty. Dubrovský et al. (2014) used an ensemble of 16 GCMs to map future drought and climate variability in the MedR. Bağçaci et al. (2021) compared the capacity of the latest release Coupled Model Intercomparison Project Phase 6 (CMIP6) model ensembles in representing near-surface temperature and precipitation of Turkey in comparison with its predecessor CMIP5 to better understand the vulnerability degree of the country to climate change. In a study conducted by Cos et al. (2022), the authors compared climate projections from CMIP5 and CMIP6 models to assess the impacts of climate change in the MedR. The findings reveal a robust and significant warming trend across all seasons, with CMIP6 models projecting stronger warming compared to CMIP5. While precipitation changes show greater uncertainties, a robust and significant decline is projected over large parts of the region during summer by the end of the century, particularly under high emission scenarios. (Seker and Gumus, 2022) uses 22 global circulation models from CMIP6 to project future precipitation and temperature changes in the MedR. The MMEs outperform individual GCMs in simulating historical data, and the projections indicate a decrease in precipitation by 15% for SSP2–4.5 and 20% for SSP5–8.5. Table 6 shows the main studies using dynamical models to forecast drought in MedR.

**Table 6 Main studies using dynamical models to forecast drought in the MedR.**

In summary, recent advancements in seasonal drought forecasting with dynamical models encompass increased climate resolution, improved representation of physical processes, improved initialization methods using data assimilation techniques (Zhou et al., 2022), use of multi-model ensembles (Wanders and Wood, 2016; Seker and Gumus, 2022), coupled modeling approaches (Guion et al., 2022), and the development of sub-seasonal to seasonal

---

[1] SSPs are the latest climate change scenarios used in CMIP6. They not only incorporate greenhouse gas emissions scenarios like their predecessor, RCPs from CMIP5, but also integrate socioeconomic factors, such as population growth, economic development, and technological progress. Essentially, SSPs provide a more holistic view of possible future climate scenarios by considering both environmental and societal changes.

predictions (Zhou et al., 2021). These steps have contributed to more accurate and reliable drought predictions. However, even with these improvements, predicting drought months in advance remains a significant challenge due to the inherent complexity and chaos of the climate system.

## 6 Hybrid Statistical-Dynamic Methods

While statistical models, when appropriately fine-tuned, can effectively predict seasonal drought events, a significant limitation arises from the non-stationary relationship between the predictors and predictands used in the forecasting process (AghaKouchak et al., 2022). This can limit their ability to accurately predict unprecedented drought events, which fall beyond the scope of their historical training data (Hao et al., 2018). On the other hand, dynamical models are proficient at capturing the nonlinear interactions among the atmosphere, land, and ocean, enhancing their ability to detect the onset of droughts (Turco et al., 2017b; Ceglar et al., 2017). However, despite their advanced capabilities, their forecast proficiency is generally constrained to a few months of lead time (Turco et al., 2017b). To address the shortcomings associated with seasonal forecasting skills, hybrid models employ statistical or machine learning methods to merge a broad variety of forecasts from statistical and dynamical models into a final probabilistic prediction product (Slater et al., 2022). The frequently used merging methods include the regression analysis, BMA, and Bayesian post-processing method (Hao et al., 2018; Strazzo et al., 2019; Han and Singh, 2020; Xu et al., 2018). The BMA method involves the estimation of the posterior probability density function (PDF) of model parameters based on the observed data and using this PDF to weight each individual model forecast (Tian et al., 2023). The hybrid forecast is then generated as the weighted average of the individual forecasts from statistical and dynamical models. The BMA weights estimation with simultaneous model uncertainty quantification can also be used in selecting the best-performing ensemble members to reduce the cost of running large ensembles (Raftery et al., 2005). There is also an opportunity to enhance the probabilistic seasonal forecast skill through Bayesian post-processing methods such as the Calibration, Bridging, and Merging (CBaM) technique (Schepen et al., 2014; Schepen et al., 2016; Strazzo et al., 2019). The calibration step consists in optimizing the dynamical forecasts from multiple GCMs by analyzing their correlation to observed data through a statistical model. In the bridging step, the dynamical forecasts from GCMs are calibrated using some large-scale climate indices (e.g., ENSO, NAO, PDO, AO), and finally, the merging component combines the forecasts of the two previous steps.

These hybrid statistical-dynamical models combine the strengths of both modeling approaches and offer several advantages compared to either statistical or dynamical models alone. Thereby, seasonal drought forecasting using hybrid models has recently become an active area of research (Madadgar et al., 2016; Strazzo et al., 2019; AghaKouchak et al., 2022). On global scale, Yan and Wood (2013) analyzed the capability of seasonal forecasting of global drought onset and found that despite climate models increasing drought detection, a significant proportion of onset events are still missed. Their findings underscore the urgent need for implementing reliable, skillful probabilistic forecasting methods to better manage the inherent uncertainties and potentially improve drought predictability. Dutra et al. (2014) confirmed that the uncertainty in long lead time forecasts suggested that drought onset might fundamentally be a stochastic problem. Mo and Lyon (2015) also found that improvements in near-real-time global precipitation observations could yield the most substantial advances in global meteorological drought prediction in the near term. This reinforces the notion that the effectiveness of dynamical models is fundamentally associated to the quality of initial data and the inherent stochastic nature of drought onset.

In line with these findings, a unique approach was undertaken by Ribeiro and Pires (2016) in the MedR. They
proposed a hybrid scheme that combines dynamical forecasts from the UK Met Office (UKMO) operational
forecasting system with past observations as predictors in a statistical downscaling approach based on MLR model
for long-range drought forecasting in Portugal (Table 7). They concluded that hybridization improves drought
forecasting skills in comparison to dynamical forecasts.
**Table 7 Main studies using hybrid statistical-dynamical models to forecast drought in the MedR.**
Leveraging these advantages of hybrid statistical-dynamical models, the prediction of flash droughts has become
possible. Indeed, these events can develop rapidly by a quick decline in soil moisture and streamflow that may
cause devastating economic and ecological impacts in a short period (from a few days to 1–2 months) (Mo and
Lettenmaier, 2015) which makes them, particularly challenging to forecast. By providing a more nuanced
understanding of the drought contributing factors, hybrid statistical-dynamical models help to identify potential
warning signs of an imminent drought event, improve drought early warning systems, and reduce false alarm rate
of drought onset (Xu et al., 2018), thus tackling some of the limitations and challenges highlighted in the earlier
studies.
**7    Discussion**
**7.1. Drought types and indices**
The indices adopted by the surveyed studies were grouped according to three distinct drought categories:
meteorological, agricultural, and hydrological. Figure 3 illustrates the percentage of usage for each index by
category. Meteorological droughts were the most common, appearing in 63.00% of the examined studies, followed
by agricultural droughts with approximately 22.20%, whereas hydrological droughts were the least prevalent,
making up only 14.80%.
The SPI was the primary indicator, used in 70.59% of meteorological drought studies. But it also served as an
indicator for hydrological and agricultural droughts, with usage rates of around 25.00% and 8.33%, respectively.
Despite the apparent versatility of the SPI, its reliance on precipitation data limits its ability to account for other
influential factors such as evapotranspiration, soil moisture, land usage, and water management practices.
Consequently, an overemphasis on the SPI could potentially constrain our comprehension of drought phenomena
in the MedR. To enrich this understanding, it is recommended to incorporate a broader range of indicators and
models that include a more diverse set of variables. Using multivariate drought indices such as the SPEI, PDSI, or
sc-PDSI, or alternatively, a combination of multiple indices, can contribute to a more comprehensive view by
including regional feedback mechanisms in the forecast process. This approach also enhances our capacity to
evaluate the impacts of global warming on drought severity and intensity in the MedR (see Marcos-Garcia et al.,
2017; Gouveia et al., 2017).
**Figure 3 Pie chart showing the proportion of use of indices in the surveyed studies in** MedR **for different drought types.**
On the other hand, SDI was the most applied index in hydrological drought studies in the MedR (37.50%). It is
calculated by comparing the current streamflow to the long-term average or median streamflow for a specific
location and time of year (Nalbantis & Tsakiris, 2009). Despite its usefulness, there are some limits to using SDI
in MedR. Indeed, this region is known for highly variable climates with strong seasonality (wet winters and dry
summers) and the presence of transient streams or intermittent rivers that flow only during and after rainfall events,
especially in sub-humid and semi-arid areas. Groundwater recharge principally occurs during the wet season, when
precipitation infiltrates the soil and replenishes aquifers (Scanlon et al., 2002). In these regions, the SDI may not
provide an accurate representation of the hydrological drought as it relies solely on streamflow data. Therefore,
the use of SDI should be done in combination with other drought indices that consider variables such as
groundwater, soil moisture, runoff, and regional variations in precipitation and streamflow patterns for accurate
hydrological drought assessment.
One can notice from Fig. 3 that the agricultural drought studies are characterized by more diversity of indices. This
diversity can be explained by the varied range of agro-climatic conditions that characterize the MedR, including a
wide range of soil types, topography, and vegetation cover. These diverse conditions can result in varying impacts
of drought on agricultural production, which require different drought indices to accurately capture the extent and
severity of the drought. In addition, the MedR is also home to a diverse range of crops, each with different
sensitivities to drought (Fereres & Soriano, 2007). This diversity of crops can require different indices to assess
the impact of drought on each crop.
Overall, a suitable index should be able to capture the impacts of drought, detect changes over time, and
differentiate between different levels of severity, while also being accurate and easily interpretable by stakeholders.
**7.2. Drought forecasting accuracy**
Key obstacles in drought modeling include the absence of a one-size-fits-all model, choosing suitable inputs,
determining an index that accurately represents drought tracking in various regions, and the uneven geographical
influence that leads to discrepancies in model accuracy (Mishra & Desai, 2005; IPCC, 2012). Consequently,
contrasting different methodologies is crucial for developing a reliable prediction model.
The accuracy of drought prediction depends on various factors such as the quality and availability of data, spatial
and temporal scales, prediction lead time, and model complexity, to cite but a few (Wilhite et al., 2014; Mishra &
Singh, 2010). For consistency, this analysis only includes studies that use $R^2$ as evaluation criteria of the forecast
with a lead time of 1 month. Joint probability models were excluded from this analysis since the accuracy
evaluation criteria were different. Moreover, the concept of lead time is not addressed in most of the surveyed
studies. It is also important to note that this analysis does not include hybrid statistical-dynamical models, as the
number of studies applying this approach in the MedR was quite limited. Consequently, the available research is
insufficient to offer a comprehensive understanding of the applicability and effectiveness of these models in the
region.
**Figure 4 Box and whiskers plot to show the performance of drought prediction models denoted by the coefficient of**
**determination ($R^2$) for the surveyed studies in MedR. The lower box shows the 25[th] percentile, the upper box shows the**
**75 percentile and the median (50[th] percentile) is represented by the black line inside the box. The whiskers show the**
**extent to the minimum and maximum values within 1.5 times the interquartile range (IQR) from the box.**
Figure 4 shows a box and whisker plot of drought forecasting model accuracy based on $R^2$ in the surveyed studies
in the MedR (see table1 in Appendix). According to the graph, hybrid models appear to be the most accurate and
consistent, with the highest median and shortest box height. Markov chains and AI models also have relatively
short box heights, indicating high agreement and accuracy across studies. Meanwhile, dynamical and regression
models exhibit moderate to high accuracy (both have median equal to 0.79), but the height of the dynamical model
box is shorter than that of the regression models, suggesting greater consistency. Time series models also show
moderate to high accuracy, with a median equal to 0.82.
Nonetheless, Fig. 4 provides valuable information about the relative performance of different models across
multiple studies in the MedR. The consistently high median of hybrid models suggests that they are particularly
effective for drought forecasting in the region. Similarly, the consistent performance of the AI and Markov chain
models, suggests that these models also show promise. The variability in the performance of the regression, and
the time series, as indicated by their taller boxplots, suggests that there may be more variability in the effectiveness
of these models across different studies and regions. The results also show that dynamical models can provide
valuable insights into drought conditions. However, the high variability in their performance, suggests that there
may be room for improvement in the development and implementation of these models in MedR.
This analysis concludes that simple statistical models such as Markov chains, regression, and time series can still
be useful in some situations and are generally more transparent and easier to interpret. For example, when focusing
on a single variable to forecast drought (e.g., precipitation using SPI), simple models like ARIMA can effectively
capture the temporal patterns and provide reasonable forecasts. Or, when drought conditions can be effectively
represented by discrete states or categories, Markov chains can be employed to model the transition probabilities
between these states and forecast future drought conditions (Habibi et al., 2018; Nalbantis and Tsakiris, 2009;
Paulo and Pereira, 2007). Also, when working with a limited number of variables and moderate interactions,
simple regression models like linear or logistic regression can provide adequate predictions of drought conditions
(Sharma et al., 2017). The effectiveness of simple models in these situations depends on the specific context and
the data quality and quantity. When more complex relationships or high-dimensional data are involved, it may be
necessary to employ more advanced models like dynamical models or combine simple models with techniques
like machine learning, copulas, or hybrid approaches to improve forecasting performance. Hybrid statistical-
dynamical models present a promising avenue for enhancing forecast accuracy, particularly for extended lead
times and in situations where intricate processes and interactions are critical (AghaKouchak et al., 2021; Mehran
et al., 2020; Madadgar et al., 2016). The relatively nascent emergence of these hybrid techniques has resulted in a
limited number of studies applying them in the MedR. This can be ascribed to factors such as data constraints,
computational complexity, and model uncertainty. Moreover, proficiency in both statistical and dynamical
modeling is needed, and interdisciplinary cooperation is frequently deficient. Notwithstanding these challenges,
there is an increasing interest not only in enhancing traditional dynamical models but also in the development and
utilization of hybrid models. As research progresses and resources become more accessible, these hybrid models
may see wider adoption for their potential to improve predictive accuracy.
### 7.3. Spatial and Temporal Scales of Drought
Figure 5 displays the spatial and temporal scales of drought forecasting studies in the MedR with a pie chart
indicating the percentage of use of drought forecasting method: statistical, dynamical, and hybrid statistical models
for each spatiotemporal scale. This figure shows that the number of droughts forecasting studies tends to decrease
as the spatial scale increases and increases as the time scale increases. We can also notice from this figure that the
majority of studies in the MedR focused on the local scales (e.g., city or catchment), particularly at annual and
seasonal time scales. In contrast, very few studies were conducted at the MedR scale, and only a few studies were
conducted at the country scale.
**Figure 5 Spatial and temporal scales of drought forecasting studies in the Mediterranean region with a pie chart**
**indicating the percentage of use of drought forecasting method: statistical, dynamical, and hybrid statistical models for**
**each spatiotemporal scale.**
When considering the spatial scale, drought forecasting becomes more challenging at larger scales due to various
factors. One of the major challenges is the complexity of the interactions between different factors that contribute
to droughts, such as precipitation, temperature, soil moisture, and vegetation cover (Sheffield & Wood, 2011).
These interactions are nonlinear and difficult to capture accurately, especially at larger scales where there are more
variability and heterogeneity (AghaKouchak et al., 2015). For instance, at the country scale, there could be
different microclimates, topography, and land use practices that affect these factors differently (Vicente-Serrano
et al., 2010a). This heterogeneity tends to increase as the spatial scale increases, making it harder to calibrate and
validate drought forecasting models. On the other hand, the small number of studies that focused on large
geographic areas is probably due to the challenge of data availability and homogeneity, which arises due to
limitations in data collection and standardization, particularly at larger spatial scales (Dai, 2011). This can lead to
incomplete or inconsistent datasets, which in turn can impact the accuracy of drought forecasting models. Remote
sensing technologies can provide a solution to this problem by allowing for the collection of large-scale, high-
resolution data that can improve the accuracy of forecasting models (Gouveia et al., 2017). The role of remote
sensing data in improving drought prediction will be further discussed in sect. 8.2.
When considering the time scale, the number of droughts forecasting studies tends to increase as the scale
increases. Drought research often emphasizes seasonal, annual, or decadal scales due to various factors. The slow-
onset nature of droughts necessitates studying their progression and recovery over extended periods (Mishra &
Singh, 2010). Investigating longer time scales also allows researchers to analyze the impact of large-scale climate
drivers, such as ENSO or NAO, on drought events (Dai, 2011). Moreover, focusing on these time scales enables
a better assessment of drought consequences on water resources, agriculture, and ecosystems, which are more
pronounced over extended periods (Wilhite & Pulwarty, 2017). Additionally, data availability and reliability tend
to be higher for longer time scales, facilitating more robust analyses. Long-term trends and climate change impacts
on droughts can also be better understood at longer time scales (Trenberth et al., 2014).
Notably, only one study focused on the weekly time scale. Drought forecasting at small scales or weekly time
scales offers several advantages, including early warning and improved water management (Pulwarty &
Sivakumar, 2014), quick response to flash droughts (Mo & Lettenmaier, 2015), support for agricultural decision-
making (Hansen et al., 2011), improved accuracy of longer-term forecasts (Yuan et al., 2015), and model
improvement and validation (Wood et al., 2016). However, drought forecasting at such a small scale may be more
challenging due to the chaotic nature of the atmosphere, making it difficult to accurately model complex
interactions between atmospheric conditions, land surface characteristics, and water management practices over
short periods (Lorenz, 1963; Seneviratne et al., 2012).
On the other hand, the most commonly used forecasting methods were statistical and hybrid statistical models,
with only a few studies applying dynamical models and the percentage of studies applying this last approach
increases with an increase in the temporal scale. There could be several reasons for these findings. Dynamical
models require large amounts of high-quality input data, which may not be readily available for the MedR due to
limitations in historical data and spatial coverage (Giorgi & Lionello, 2008). Statistical and hybrid statistical
models often have lower data requirements and are generally computationally more efficient than dynamical
models, making them more suitable for regions with limited data availability and computational constraints.
Furthermore, the percentage of studies applying dynamical models increases with an increase in the temporal scale
because these models are better suited for capturing long-term climate variability and the influence of large-scale
climate drivers (Dai, 2011; Sheffield et al., 2012). Statistical and hybrid statistical models, conversely, are more
effective at capturing short-term variability and local-scale processes, which are often more relevant for drought
forecasting in the MedR (Mehran et al., 2014). Lastly, data availability at shorter temporal scales can be a limiting
factor for developing and validating dynamical models (Shah et al., 2018).
In summary, while increasing the spatial scale can decrease the accuracy of drought forecasting studies, increasing
the time scale can improve the accuracy by allowing for a more comprehensive understanding of the various factors
that contribute to drought conditions. It is essential to consider both spatial and temporal scales when conducting
drought forecasting studies to ensure the most accurate predictions possible.
**8   Challenges and Future Prospects**
In the earlier discussion, we analyzed drought indices, factors affecting the accuracy of drought forecasts, and the
significance of spatial and temporal scales in drought predictions within the MedR context. Building on this
understanding, the following sections will focus on the challenges and prospects within the realm of drought
forecasting, which will help to pinpoint potential avenues for progress and innovation in this area.
**8.1. Data Assimilation**
The lack of in-situ measurement networks and coarse global seasonal forecast skills
has hindered drought forecasting facilities, especially in data-poor regions (Pozzi et al.,
2013; Haile et al. 2020). In this regard, Data Assimilation (DA) provides a powerful approach to enhancing drought
forecasting accuracy by incorporating different observations and climate forecasts into a hydrologic model to
generate more precise initial conditions (Hao et al., 2018; Tang et al., 2016). Therefore, many studies have referred
to this method to better forecast hydroclimatic variables (e.g., Bazrkar and Chu, 2021; Peng, 2021; Xu et al., 2020;
Liu et al., 2019; Steiger et al., 2018; Steiger and Smerdon, 2017). The ensemble Kalman Filter (EnKF) (Evensen,
1994) algorithm is one of the most popular DA techniques applied by the hydrologic community. However, this
assimilation method is subject to some inherent drawbacks especially in nonlinear dynamic systems thus resulting
in suboptimal performance and violation of water balance (Abbaszadeh et al., 2018). Given these limitations,
emphasis should be placed on the development of improved DA algorithms better adapted to hydrologic models,
which allow the modeling of different temporal and spatial scales and the improvement of water balance. This can
be achieved by modifying the standard approaches such as the ensemble Kalman filter or variational algorithms
so that, accurate predictions can be obtained at a reasonable computational cost. These include among others hybrid
EnKF-Var methods (Bannister, 2017; Bergou et al., 2016; Mandel et al., 2016) and AI algorithms for ensemble
post-processing (Grönquist et al., 2021). One recent advance in data assimilation techniques for drought
forecasting is the use of machine learning algorithms to improve the accuracy of predictions. For example,
researchers have used machine learning techniques to develop models that can analyze large amounts of data from

a variety of sources and generate more accurate forecasts of drought conditions (Aghelpour et al., 2020; Rhee and Im, 2017; Feng et al., 2019). These models can also be updated in real time as new data becomes available, allowing for more accurate and up-to-date forecasts. Another advance in data assimilation techniques for drought forecasting is the use of remote sensing data and reanalysis to improve the accuracy of predictions, which may be particularly beneficial in areas where ground-based observations are limited (Shahzaman et al., 2021b; Shi et al., 2011).

## 8.2. Remote Sensing and Reanalysis

Various challenges in drought modeling in the MedR are related to data availability. The lack of climatic and hydrological observations in ungauged catchments, low station density, short data records, data gaps, and limited data access in some Mediterranean countries. All these challenges can limit the accuracy and reliability of drought predictions. Although many efforts are being deployed by developing new complete datasets in the MEDR (Tuel and El Moçayd, 2023), finding alternative data sources and modeling techniques is essential to tackle these challenges.

Remote sensing data can provide real-time information about the Earth's surface facilitating effective drought forecasting, monitoring, and early warning (Zhang et al., 2016). Agricultural drought can be assessed by analyzing changes in vegetation cover over time. Indeed, drought can lead to marked changes in the health and vigor of vegetation, and these changes can be detected using remote sensing data (Belal et al., 2014). By analyzing changes in vegetation greenness over time, it is possible to identify areas that are experiencing or are at risk of experiencing drought stress. Moreover, drought conditions related to vegetation or evapotranspiration can also be monitored with drought indices from remote sensing products, such as NDVI or Evaporation Stress Index (ESI) (Shahzaman et al., 2021a). Microwave satellite data can also be used to estimate soil moisture levels during crop growing season, which can be used to predict and monitor potential agricultural droughts (Le Page and Zribi, 2019; Yuan et al., 2015).

In addition, satellite observations of precipitation and soil moisture such as IMERG (Huffman et al., 2015), PERSIANN-CCS (Sadeghi et al., 2021), CHIRPS (Funk et al., 2015), SMAP (Entekhabi et al., 2010), MSWEP V2 (Beck et al., 2019), GLEAM v3 (Martens et al., 2017), and DROP (Turco et al., 2020) can be used in conjunction with the in-situ observations and ground-based radar observations data to fill observational gaps.

Moreover, data from numerical weather forecasting reanalysis such as ERA5-land were used instead or along with direct observations to forecast drought in many studies (Babre et al., 2020; Junqueira et al., 2022; Parker et al., 2021). ERA5-land is a state-of-the-art global reanalysis dataset that can provide a consistent view of the evolution of land variables (e.g., precipitation, temperature) over several decades at an enhanced resolution (~10 km). This product obtained by assimilating observations through a 4D-VAR data assimilation technique can be used as ground truth in data-poor regions. For example, ERA5-land can be used to calibrate and validate climate forecasts and to choose an ensemble of the most skilled GCMs in reproducing the actual observed climate in a specific region.

Similarly, SAFRAN, a high-resolution meteorological reanalysis, has shown its utility in regions with sparse observational data. Tramblay et al. (2019) used SAFRAN to generate a high-resolution (5 km) gridded daily precipitation datasets for Tunisia between 1979 and 2015. Their study, which combined data from 960 rain gauges

with the SAFRAN analysis, demonstrated that SAFRAN surpassed other standard interpolation methods like
Inverse Distance, Nearest Neighbors, Ordinary Kriging, or Residual Kriging with altitude. The outcome was a
highly accurate gridded precipitation dataset that could be instrumental for climate studies, model evaluation, and
hydrological modeling to support the planning and management of surface water resources.
Finally, remote sensing data and reanalysis remain valuable tools for drought forecasting and monitoring, as it
provides timely land surface information that can fill the observational gaps, help to identify areas at risk of
potential drought conditions and to monitor the progression of drought over time.

### 8.3. Uncertainty analysis in drought forecasting

In spite of the large number of studies that have been carried out on the probabilistic characterization of drought,
the quantification of uncertainty of these forecasts is still ignored in major studies. Uncertainty analysis is an
important aspect of probabilistic drought forecast, as it allows users to understand the degree of confidence
associated with the forecasted probabilities (Hao et al., 2016; Dehghani et al., 2014). Therefore, more efforts
should focus on quantifying the uncertainty beyond just an ensemble of model simulations (AghaKouchak et al.,
1012 2022).

Drought forecasting is subject to epistemic and aleatory uncertainties. The first one arises from incomplete
knowledge of drought processes and can be reduced with improved understanding, more data, and good models'
calibration and validation. The second one is related to the inherent variability and randomness in natural systems
and is often difficult to reduce (Pappenberger & Beven, 2006). In addition, uncertainties in drought forecasting
can vary by region, spatial scale, and temporal scale. As we discussed in sect. 7.3, even well calibrated and
validated, the drought forecasting model will not necessarily perform equally well in all periods or locations. By
considering the uncertainty of the drought model as a nonstationary process in space and time, researchers can
gain new insights into the variability of uncertainty and its underlying causes (AghaKouchak et al., 2022). This
perspective can help identify regions or periods where the uncertainties are particularly high, which can guide
further research, data collection, and model development efforts. Additionally, understanding the space-time
variability of uncertainty can inform the development of more robust and reliable forecasting and decision-making
approaches that account for the changing nature of uncertainty.
Various techniques can be employed to quantify drought forecast uncertainty, including ensemble forecasting
(Palmer et al., 2004), Bayesian methods (Vrugt et al., 2008), sensitivity analysis (Saltelli et al., 2008) and
probabilistic forecasting (Gneiting et al., 2005). Probabilistic drought prediction can also involve the use of data
assimilation techniques to integrate different data sources, including remote sensing data, ground-based
observations, and output from meteorological and hydrological models. Lately, hybrid statistical-dynamical
models have shown their potential in reducing uncertainties associated with both statistical and dynamical methods
(Yuan et al., 2015; Madadgar et al., 2016). For example, shortcomings in dynamical model physics or data can be
counterbalanced by the empirical associations in statistical models. While, uncertainties in statistical models
resulting from shifting climate conditions can be tackled by the physically based dynamical models (Yuan et al.,
1034 2015).

In summary, probabilistic drought prediction with uncertainty analysis can be useful tools for decision makers, as
they provide a more comprehensive view of the potential impacts of drought and allow for more informed risk

management decisions. However, what is missing in the current drought forecasting models is not just the uncertainty quantification, but also a lack of awareness of it (AghaKouchak et al., 2022).

### 8.4. Drought Information Systems

A critical component of proactive approaches to drought preparedness is providing timely and reliable climate information, including seasonal forecasts, that helps decision makers prepare management policies (Manatsa et al., 2017). Identifying drought risk timely depends on our ability to monitor and forecast its physical causing mechanisms at the relevant spatiotemporal scale. An integrated national drought monitoring and early warning system has been implemented in many regions and countries such as the United States, New Zealand, South Asia, India, and Europe (Prabhakar and Rama, 2022) but has not taken place until recently in developing countries (e.g., the Southern and Eastern Mediterranean countries). This is probably due to the lack of a drought information system, the sparse observation networks, and the low predictability of seasonal precipitation in these countries. To overcome these limitations, there is a need for developing a Drought Information System with a complete approach allowing data collection and preprocessing, accurate probabilistic drought risk prediction using a combination of ensemble climate seasonal forecasts, ground-based observations, reanalysis, conventional and remote-sensing observations, artificial intelligence, data assimilation and hydrological models and drought information dissemination through a web-based Drought Early Warning System (DEWS).

## 9 Conclusions

This study reviewed the recent statistical, dynamical, and hybrid statistical-dynamical methods used to forecast droughts and their application on the MedR. Drought definitions, classification, indices, and causative physical mechanisms were also presented in the context of the MedR. The main conclusions of this review are:

1. There are only a few studies on the analysis of physical mechanisms causing droughts in the MedR. The review of these studies confirmed that seasonal drought predictability skills are still very limited over the region due to its relatively poor teleconnection with ENSO compared to the tropical and subtropical regions. Besides, MedR is strongly influenced by other climate patterns, such as the NAO, regional MO, ULMOi, and NAWA which can also affect the region's weather and climate but their relationship to drought onset is rather weak and could not explain major droughts in the region. Land surface memory can also contribute to the predictability of seasonal and sub-seasonal droughts. Thereby, an accurate representation of these land-atmosphere processes is needed to improve drought forecasting skills in mid-latitude regions such as the Mediterranean.
2. Statistical models were largely used to forecast droughts in the MedR. One of the major limitations of these models is that they often assume a stationary relationship between the predictors and the predictands which can lead to potentially inaccurate forecasts. In this regard, AI models such as SVR, SVM, and ANN have proven good capacity in detecting local discontinuities and non-stationary characteristics of the data and show satisfactory forecasting skills at less than 6 months lead time. Moreover, sophisticated statistical models, incorporating a data pre-processing technique such as wavelet analysis, EMD, or PCA with AI models have proven to be more efficient than using a single model and can extend the lead time of the drought forecast up to 12 months. The copulas can also provide valuable insights into the complex

relationships between different drought predictors. The use of copulas enables a more in-depth analysis of the nonlinear dependencies between variables such as temperature, precipitation, and soil moisture, yielding a more comprehensive understanding of the factors that contribute to drought risk in a specific region. This leads to a more sophisticated and reliable forecast of drought probability. Thus, copulas are a highly useful resource in the ongoing effort to understand and manage the consequences of drought.

3. Dynamical models, given their ability to capture nonlinear interactions across the atmosphere, land, and ocean, offer considerable potential for more accurate and reliable seasonal drought predictions. However, the inherent chaotic nature of the atmosphere restricts their forecast skill to a few months in advance. The dynamical drought forecasting has seen notable advancements, such as enhanced climate model resolution, refined representation of physical processes, improved initialization methods, the application of multi-model ensembles, and the development of coupled modeling approaches. These developments have indeed bolstered the accuracy and reliability of drought predictions. Nevertheless, the implementation of these models in the MedR is constrained by challenges such as limited data availability, computational complexity, and inherent model uncertainties.

4. Hybrid statistical-dynamical models can be promising tools to potentially enhance the accuracy and reliability of drought forecasting in the MedR. By merging a broad variety of forecasts from statistical and dynamical models into a final probabilistic prediction, hybrid models benefit from the strengths of both modeling approaches and improve the forecast skill compared to an individual model. But their applicability remains challenging due to several constraints. Indeed, the hybrid model may require careful calibration and validation to ensure that they are performing optimally which can be time-consuming, requiring a large amount of data, specialized expertise, and high computational resources.

5. One of the major challenges in drought forecasting in the MedR is the lack of long-term, high-quality hydroclimatic observations to convey the nonstationary patterns and the variability of the climate. In addition, hydrologic model predictions are often poor, due to model initialization, parametrization, and physical errors. To address these challenges, it is important to improve the availability and quality of data for drought forecasting in this region. This could involve implementing better monitoring systems and increasing the number of weather stations in the region. In addition, efforts should be made to improve the performance of drought forecasting models by using more advanced data assimilation and machine learning techniques and to incorporate data from other sources such as state-of-art satellite observations and reanalysis with relatively high spatiotemporal analysis to provide a superior hydrologic and climate states estimate and consequently a skillful agricultural and hydrological drought forecasting.

6. Drought mapping is the final stage in which drought risk information is disseminated and communicated to end users. Major studies in the MedR analyze drought risk using some drought indices without applying a visualization via maps or presenting the risk on a single map showing the overall risk situation. An informative visualization of results via probabilistic drought risk maps is recommended, whereby color gradations or contouring are used to effectively illustrate the range of probabilities. Ensuring cartographic rigor, such maps should maintain spatial accuracy, use appropriate scaling, and include a clearly defined legend to decrypt different probability levels. Uncertainties related to drought modeling and prediction also need to be perspicuously defined, discussed and communicated to increase the intelligibility and comprehensibility of decision makers, farmers, and other end users.

7.  Finally, much effort should be done to improve the communication and dissemination of drought forecasts

1116        which can help in extending their lead time by ensuring that decision makers and stakeholders have access

1117        to the most up-to-date information.

**Index of Acronyms**

Adaptive neuro-fuzzy inference systems (ANFIS)
Akaike's Information Criterion (AIC)
Anderson-Darling (AD)
Artificial neural network of multilayered perceptron (ANN-MLP)
Asymmetric Power Autoregressive Conditional Heteroskedasticity (APARCH)
Atmospheric water deficit (AWD)
Automated Statistical Downscaling (ASD)
AutoRegressive (AR)
Autoregressive Conditional Heteroskedasticity time series of order 1 (ARCH)
Autoregressive integrated moving average (ARIMA)
Autoregressive moving average (ARMA)
Autoregressive moving average time series of order (11) (ARMA)
Autoregressive moving average time series of order 1 (MA1)
Autoregressive moving average time series of order 2 (MA2)
Autoregressive time series of order 1 (AR1)
Autoregressive time series of order 2 (AR2)
Bagging (BG)
Bagnouls-Gaussen aridity index (BGI)
Bayesian Information Criterion (BIC)
Breaks for Additive Season and Trend (BFAST)
Coefficient of efficiency (CE)
Convolutional neural network long short-term memory (CNN-LSTM)
Co-ordinated regional climate downscaling experiment for the Mediterranean area (MedCORDEX)
Corrected and unbiased trend-free-pre-whitening (TFPWcu)
Coupled Model Intercomparison Project (CMIP)
Cramers-von Mises (CvM)
Crop moisture index (CMI)
Drought class transition probabilities (DCTP)
Empirical Mode Decomposition (EMD)
Exponential General Autoregressive Conditional Heteroskedasticity time series of order (11)) (EGARCH)
False alarm ratio (FAR)
Frequency bias (FB)
Generalized Autoregressive Conditional Heteroskedasticity time series of order (11) (GARCH)
Geometric Brownian Motion (GMB)

Geometric Brownian Motion time series model with asymmetric Jumps (GBMAJ)
Global Historical Climatology Network-Monthly (GHCN)
Global Land Data Assimilation System (GLDAS)
Groundwater Resource Index (GRI)
Growing season minimum and maximum values (gsmm)
Hadley Centre Coupled Model version 3(HadCM3)
Kolmogorov-Smirnov (K-S)
Land Surface Temperature (LST)
Maximum likelihood methods (MLIKE)
Mean absolute error (MAE)
Mean error (ME)
Model output statistics (MOS)
Moderate Resolution Imaging Spectroradiometer (MODIS)
Modified Fournier Index (MFI)
Monthly average relative humidity (MARH)
Monthly mean solar radiation (MMSR)
Moving average (MA)
Multiple Linear Regression (MLR)
National Center for Atmospheric Research (NCAR)
National Centers for Atmospheric Prediction (NCEP)
National Oceanic and Atmospheric Administration (NOAA)
NDVI anomaly index (NDVIA)
Non-linear AutoRegressive with eXogenous inputs (NARX)
Normalized Difference Vegetation Index (NDVI)
Normalized Difference Water Index (NDWI)
North Atlantic Oscillation (NAO)
Pedotransfer functions (PTF)
Periodic autoregressive (PAR)
Periodic autoregressive moving average (PARMA)
Principal component analysis (PCA)
Probability of detection (POD)
Probability of false detection (POFD)
Proportion of correct predictions (PC)
Random forest (RF)
Random subspace (RSS)
Random tree (RT)
Reconnaissance Drought Index (RDI)
Root mean squared error (RMSE)
Sea Surface Temperature (SST)
Seasonal-ARIMA (SARIMA)
Soil and Terrain Database (SOTER)

Soil Moisture (SM)
Soil Moisture Agricultural Drought Index (SMADI)
Soil Moisture and Ocean Salinity (SMOS)
Soil moisture anomaly index (SMAI)
Soil Moisture Deficit Index (SMDI)
Soil moisture percentiles (Wp)
Soil Water Deficit Index (SWDI)
Soil Wetness Deficit Index (SWetDI)
Standardized Water-Level Index (SWI)

Streamflow drought index (SDI)
Support vector Regression (SVR)
Temperature Condition Index (TCI)
The Second Generation of Canadian Coupled General Circulation Model (CGCM2)
Vegetation Condition Index (VCI)
Vegetation Health Index (VHI)
Wavelet Analysis (WA)
Wavelet decomposition (WD)


## Competing Interests

The authors declare that they have no conflict of interest.

## Author contribution

Each author has made substantial contributions to the creation of this manuscript. BZ was responsible for
conceptualization, methodology, investigation, analysis, drafting the manuscript, and reviewing and editing. NEM
contributed to the methodology, analysis, review, and editing processes. EHB was involved in the methodology,
analysis, review, and editing stages.
**Disclaimer**
**Acknowledgments**

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

**Table 1** Main studies using the Time series model to forecast drought in the MedR.

| Reference | Inputs | Outputs | Methods | Time scale | Study area | Drought type | Study period |
|---|---|---|---|---|---|---|---|
| (Bouznad et al., 2021) | Precipitation, temperature, and ET | Aridity index, SPI, NDVI | ARIMA, SARIMA | Monthly, annual | Algeria | Meteorological | Baseline 1985–2014 Future 2015–2024 |
| (Achite et al., 2022) | Monthly precipitation | SPI12, SRI12 | ARIMA, SARIMA | Annual | Algeria | Meteorological, hydrological | 1972–2018 |
| (Al Sayah et al., 2021) | LANDSAT imageries at a 3-year interval, and meteorological indicators | MFI, BGI, VHI, VCI, TCI, NDWI, NDVI | ARIMA/SARIMA | Annual | Lebanon | Meteorological, hydrological and agricultural | 1990–2018 |
| (Tatli, 2015) | IPCC observed precipitation | PDSI | Hurst exponent, Mann - Kendall test | Monthly | Turkey | Meteorological | 1966–2010 |
| (Pablos et al., 2017) | LST, NDVI  Satellite SM data (SMOS BEC L4 and MODIS SR) and In Situ SM Data | SWDI, SMADI, SMDI, SWetDI, AWD CMI | POD; POFD; FAR; FB | Weekly | Spain | Agricultural | 2010- 2016 |
| (Hadri et al., 2021) | NDVI ; Rainfall | SPI, SWI | The Mann-Kendall and Sen's slope | Seasonal | Morocco | Meteorological, agricultural | 2008-2017 |
| (Ben Abdelmalek and Nouiri, 2020) | Monthly rainfall series in 16 main meteorological stations | SPI, RDI, Annual PET | Mann - Kendall test, Weighted Inverse Distance interpolation | Annual | Tunisia | Meteorological, agricultural | 1973–2016 |
| (Karabulut, 2015) | Precipitation | SPI | Cumulative Deviation Curve | Monthly, seasonal, annual | Turkey | Meteorological | 1975–2010 |
| (Jiménez-Donaire et al., 2020) | Rainfall, soil moisture, and vegetation (NDVI) | SPI, NDVIA  SMAI | Combined Drought Indicator | Monthly, seasonal, annual | Spain | Agricultural | 2003–2013 |
| (Ben Mhenni et al., 2021) | SM (SOTER); MedCORDEX daily grided reanalysis of meteorological data; NOAA weekly NDVI | SPI, SPEI, PDSI, and Wp | Lag-correlation analysis | Seasonal, annual | Tunisia | Meteorological, agricultural | 1982–2011 |
| (Derdous et al., 2021) | Rainfall | SPI | the Mann–Kendal, Sen's slope estimator, and the Pettitt test; | Monthly, seasonal, annual | Algeria | Meteorological | 1936 –2008 |

| | | | | | | | |
|---|---|---|---|---|---|---|---|
| (Mendes et al., 2022) | Precipitation, water level in reservoirs | SPI14 | BFAST | Seasonal | Portugal | Hydrological | 1978-2020 |


**Table 2** Main studies using regression analysis to forecast drought in the MedR

| Reference | Inputs | Outputs | Methods | Time scale | Study area | Drought type | Study period |
|---|---|---|---|---|---|---|---|
| (Sousa et al., 2011) | Monthly rainfall  SST, NAO | PDSI, scPDSI | Calibrated Stepwise Regression | Monthly, seasonal, annual | MedR | Meteorological | 1901–2000 |
| (Papadopoulos et al., 2021) | Monthly precipitation | SPI, RDI | Fuzzy linear regression analysis | Monthly, seasonal, annual | Greece | Meteorological | 1996–2016 |
| (Martínez-Fernández et al., 2016) | In situ hourly SM, daily rainfall, daily PET, and SMOS data | SWDI | PTF; linear regression | Weekly, Seasonal | Spain | Agricultural | 2010–2014 |
| (Tigkas and Tsakiris, 2015) | Monthly rainfall; average monthly mean, max, and min temperature | PET, RDI | Multiple regression models | Monthly, seasonal, annual | Greece | Agricultural | 47-50 years |


**Table 3** Main studies using Artificial Intelligence Models to forecast drought in the MedR.

| Reference | Inputs | Outputs | Methods | Time scale | Study area | Drought type | Study period |
|---|---|---|---|---|---|---|---|
| (Mohammed et al., 2022) | Precipitation | SPI | BG, RSS, RT, and RF | Monthly, seasonal, annual | Syria | Agricultural, Hydrological | 1946-2005 |
| (Di Nunno et al., 2021) | Precipitation and discharge | | NARX neural networks | Seasonal | Italy | Hydrological | 1997-2020 |
| (El Aissaoui et al., 2021) | Monthly average precipitation; Monthly min/max air temperature; MARH; MMSR | SPI, SPEI | SVR1: linear; SVR2: Polynomial; SVR3: RBF; SVR4: sigmoid | Monthly | Morocco | Meteorological | 1979–2013 |
| (Achour et al., 2020) | Monthly rainfall data | SPI | TFPWcu; ANN | Monthly, seasonal and annual | Algeria | Meteorological | 1960–2010 |

| Reference | Inputs | Outputs | Methods | Time scale | Study area | Drought type | Study period |
|-----------|--------|---------|---------|-----------|-----------|--------------|--------------|
| (El Alaoui El Fels et al., 2020) | Monthly rainfall amount | SPI | PCA, Frequency analysis, ANN | Monthly, annual | Morocco | Meteorological | 1970–2017 |
| (El Ibrahimi and Baali, 2018) | Observed SPI | Predicted SPI | ANFIS; ANN-MLP; SVR, ANN, WA-ANFIS WA-SVR, WA-ANN-MLP | Monthly, seasonal, annual | Morocco | Meteorological | 1978–2014 |
| (Djerbouai and Souag-Gamane, 2016) | Historical monthly rainfall | SPI | ARIMA, SARIMA, WA-ANN | Monthly, seasonal, annual | Algeria | Meteorological | 1936–2008 |
| (Myronidis et al., 2012) | Monthly precipitation Monthly in-situ measurements of water lake levels | SPI | ARIMA-ANN | Annual and seasonal | Greece | Meteorological | 1973–2008 |
| (Danandeh Mehr et al., 2022) | Rainfall and temperature time series | SPEI-3 and SPEI-6 | CNN-LSTM, genetic programming, ANN, LSTM and CNN | Monthly | Turkey | Meteorological | 1971–2016 |
| (Başakın et al., 2021) | Monthly sc-PDSI | Predicted sc-PDSI | ANFIS, EMD-ANFIS | Monthly, seasonal, | Turkey | Meteorological | 1900–2016 |
| (Özger et al., 2020) | Monthly sc-PDSI | Predicted sc-PDSI | EMD, WD, ANFIS, SVM, WD-ANFIS, EMD-ANFIS, WD-SVM, | Monthly, seasonal | Turkey | Meteorological | 1900–2016 |


**Table 4** Main studies using Joint Probability Models to forecast drought in the MedR.

| Reference | Inputs | Outputs | Methods | Time scale | Study area | Drought type | Study period |
|-----------|--------|---------|---------|-----------|-----------|--------------|--------------|
| (Bouabdelli et al., 2020) | Monthly precipitation, temperature 9 GCMs of CMIP5 | SPI12, SDI6 | Copula theory, Hydrological modeling using GR2M | Seasonal, annual | Algeria | Meteorological, Hydrological | Baseline: 1941–2011, Future: 2021–2100 |
| (Bonaccorso et al., 2015) | NAO; areal monthly precipitation series; | SPI | DCTP (SPI, NAO) | Monthly, seasonal | Sicily, Italy | Meteorological | 1921–2008 |
| (Serinaldi et al., 2009) | Mean areal precipitation, aggregated at 6 months | SPI; joint return periods of drought | Probabilistic analysis of drought characteristics using Copula | Seasonal | Italy | Meteorological | 1921–2003 |
| (Hamdi et al., 2016) | Daily streamflow data, | The joint probabilities and bivariate | Two-dimensional copula model; | Annual | Tunisia | Hydrological | 1966–2008 |

| | | | return periods | | the threshold level method | | | | |
|---|---|---|---|---|---|---|---|---|---|

| (Esit and YUCE, 2022) | Monthly precipitation | SPI | Two-dimensional copula model | Seasonal | Turkey | Meteorological | 1963–2016 |
|---|---|---|---|---|---|---|---|
| (Tosunoglu and Can, 2016) | Monthly rainfall series | SPI; probabilistic properties of droughts | Two-dimensional copula model | Monthly | Turkey | Meteorological | 1966–2006 |


**Table 5** Main studies using Markov Chains Model to forecast drought in the MedR.

| Reference | Inputs | Outputs | Methods | Time scale | Study area | Drought type | Study period |
|---|---|---|---|---|---|---|---|
| (Habibi et al., 2018) | Annual precipitation from 65 meteorological stations | SPI | Markov chain models, DI and 11 time series models (GMB, GBMAJ, APARCH, AR1, AR2, ARCH, ARMA, EGARCH, GARCH, MA1, MA2) | Annual | Algeria | Meteorological | 1960–2010 |
| (Paulo and Pereira, 2007) | 67-year averages of monthly precipitation | SPI | Non-homogeneous and homogeneous Markovian modeling | Monthly, seasonal, annual | Portugal | Meteorological | 1931/32 – 1998/99 |
| (Lazri et al., 2015) | Annual precipitation maps from meteorological satellite data; 219 rain gauges and radar precipitation | SPI | Markov chain model; Transition probability matrix | Annual | Algeria | Meteorological | 2005–2010 |
| (Nalbantis and Tsakiris, 2009) | Monthly Precipitation, monthly streamflow | SPI, SDI | Non-stationary Markov chain | Monthly, seasonal, annual | Greece | Hydrological | 1970–71 to 1999–2000. |
| (Akyuz et al., 2012) | Observed annual streamflow | Probabilities and return periods of droughts | First-order Markov chain model, second-order Markov chain model | Annual | Turkey, New work, Sweden | Hydrological | 1938–2005 |
| (Cancelliere et al., 2007) | Monthly Precipitation in 43 precipitation stations | SPI | Markov chain model | Seasonal, annual | Sicily, Italy | Meteorological | 1921–2003 |


**Table 6** Main studies using dynamical models to forecast drought in the MedR.

| Reference | Inputs | Outputs | Methods | Time scale | Study area | Drought type | Study period |
|---|---|---|---|---|---|---|---|

| Reference | Data | Drought Index | Method/Model | Time scale | Region | Drought type | Period |
|---|---|---|---|---|---|---|---|
| (Elkharrim and Bahi, 2014) | Historical precipitation; HadCM3(monthly precipitation and temperature); Observed GHCN v3; NCEP and NCAR reanalysis | SPI | ASD | Seasonal and annual | Morocco | Meteorological | Baseline 1961-2010 Future 2014-2099 |
| (Marx et al., 2018) | GCMs: GFDL-ESM2M, HadGEM2-ES, IPSL-CM5A-LR, MIROC-ESM-CHEM, NorESM1-M | | Hydrological models: mHM, Noah-MP, and PCR-GLOBWB | Annual | Europe | Meteorological and hydrological | Baseline 1971–2000 |
| (Vasiliades and Loukas, 2009) | Observed runoff | PDSI, Weighted PDSI, PHDI and the moisture anomaly Z-index; runoff and soil moisture | monthly UTHBAL conceptual water balance model | Monthly | Greece | Meteorological and hydrological | 1960–2002 |
| (Brouziyne et al., 2020) | CNRM-CM5 (RCP4.5, RCP8.5); GLDAS 25 km reanalysis data; Observed daily rainfall and temperature (max and min) series | SPI-12; SDI-12; Monthly runoff, rainfall; Future water yield. | Hydrological model SWAT; | Annual | Morocco | Meteorological, Hydrological | Baseline 1985-2005; Future 2030–2050 and 2080–2100 |
| (Mendicino et al., 2008) | Monthly precipitation, temperature, SPI, NDVI | GRI | A water balance model | Seasonal, annual | Italy | Meteorological and Hydrological | 1959–2006 |
| (Dubrovský et al., 2014) | Monthly and daily precipitation and temperature outputs from 16 GCMs simulations (IPCC-AR4) | PDSI, Z-index | Multi-GCM forecast | Seasonal | MedR | Meteorological | Baseline 1961–1990; Future 2070–2100 |
| (Ruffault et al., 2014) | Daily precipitation, temperature and global radiation from ARPEGE-Climate model Version 4; Historical observations from SAFRAN dataset | Maps of summer precipitations, number of wet days in summer and drought intensity | Water balance model, quantile mapping/anomaly method | Annual seasonal | France | Agricultural, Hydrological | Baseline 1961–1990 Future 2071–2100 |


**Table 7** Main studies using hybrid statistical-dynamical models to forecast drought in the MedR.

| Reference | Inputs | Outputs | Methods | Time scale | Study area | Drought type | Study period |
|---|---|---|---|---|---|---|---|
| (Ribeiro and Pires, 2016) | UKMO operational forecasting system | SPI3 | MLR | Seasonal, annual | Portugal | Meteorological, agricultural, and hydrological | 1987–2003 |

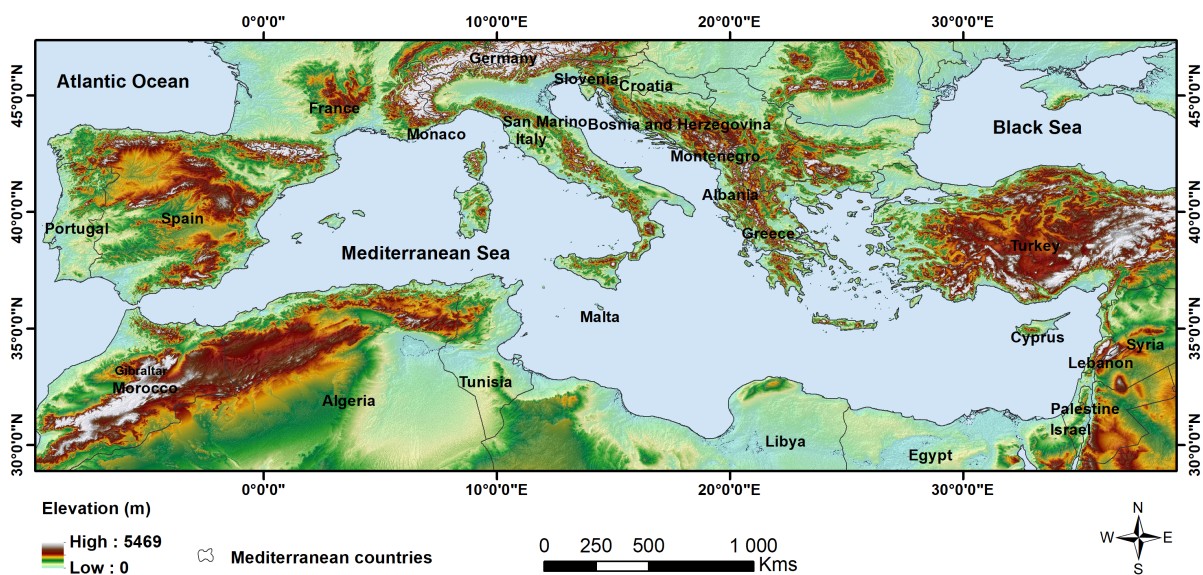

**Figure 1 Topography of the Mediterranean Region** (30°N - 46°N in latitude and 10°W - 40°E in longitude)**.**

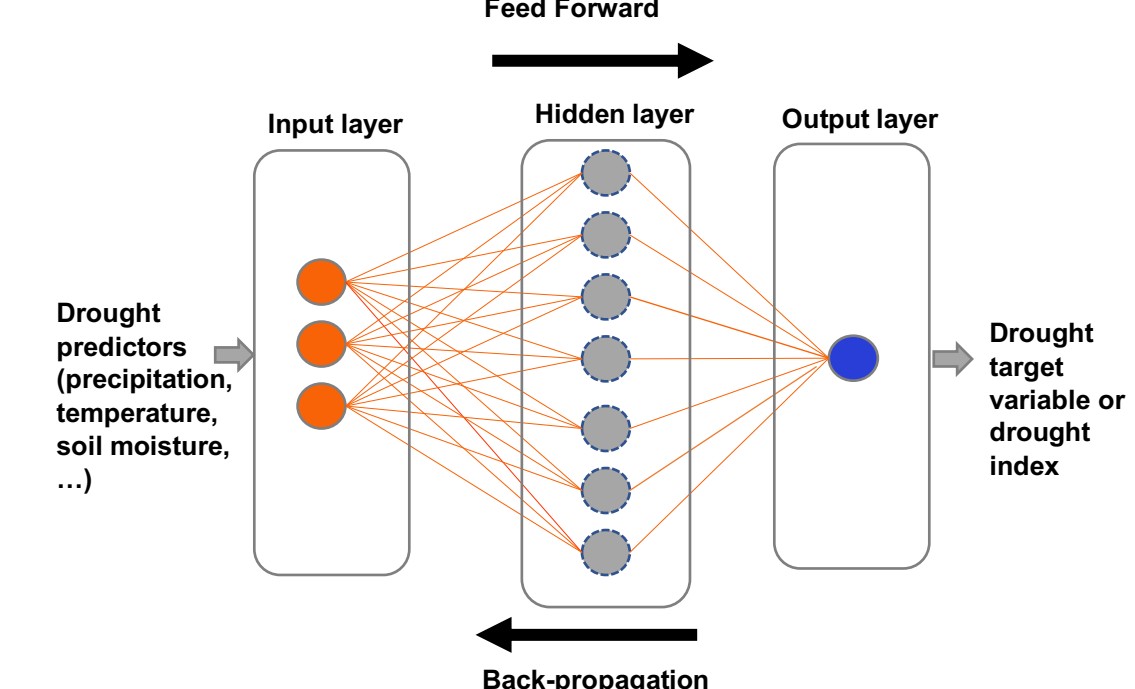

**Figure 2 Drought forecasting based on a simple ANN architecture.**


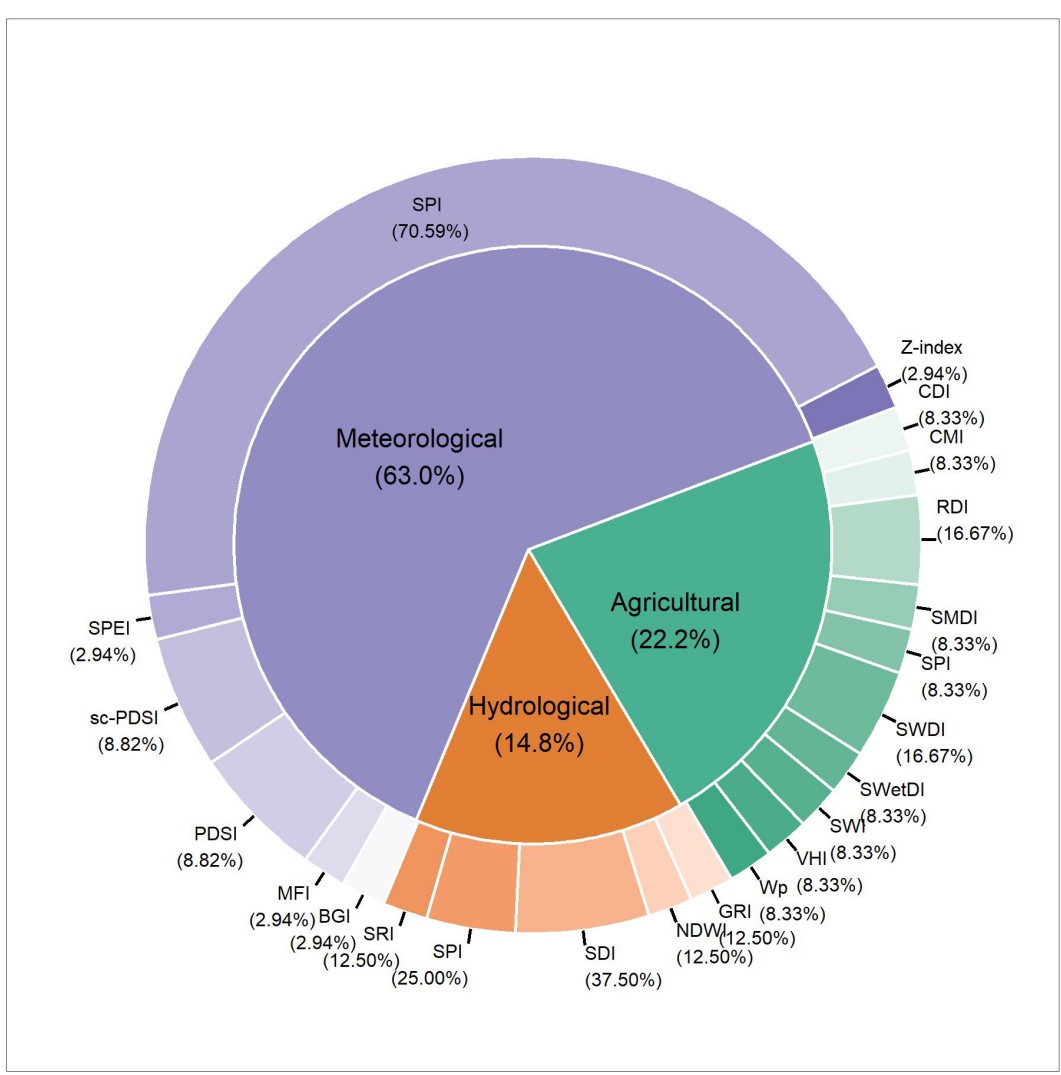


**Figure 3 Pie chart showing the proportion of use of indices in the surveyed studies in MedR (Tables 1-7) for different**
**drought types.**

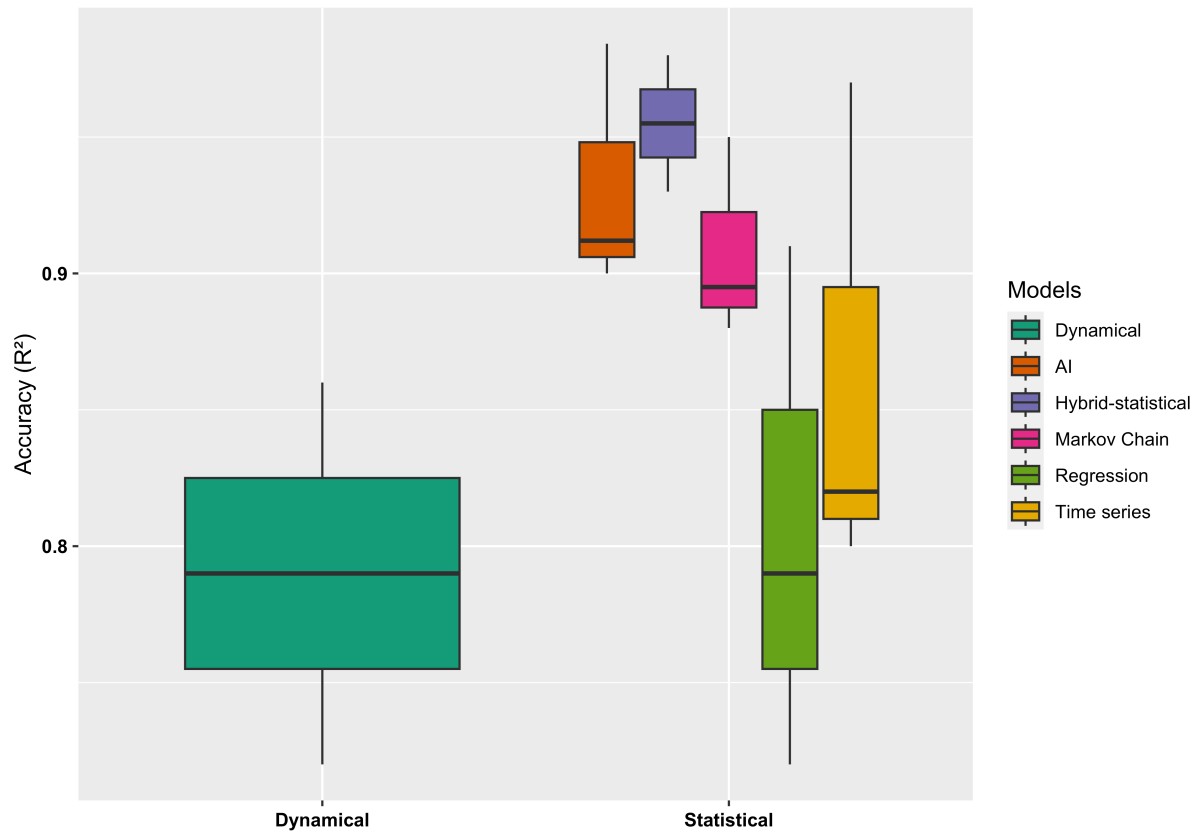


**Figure 4 Box and whiskers plot showing the performance of drought prediction models denoted by the coefficient of determination (R²) for the surveyed studies in MedR. The lower box shows the 25th percentile, the upper box shows the 75 percentile and the median (50th percentile) is represented by the black line inside the box. The whiskers show the extent to the minimum and maximum values within 1.5 times the interquartile range (IQR) from the box.**


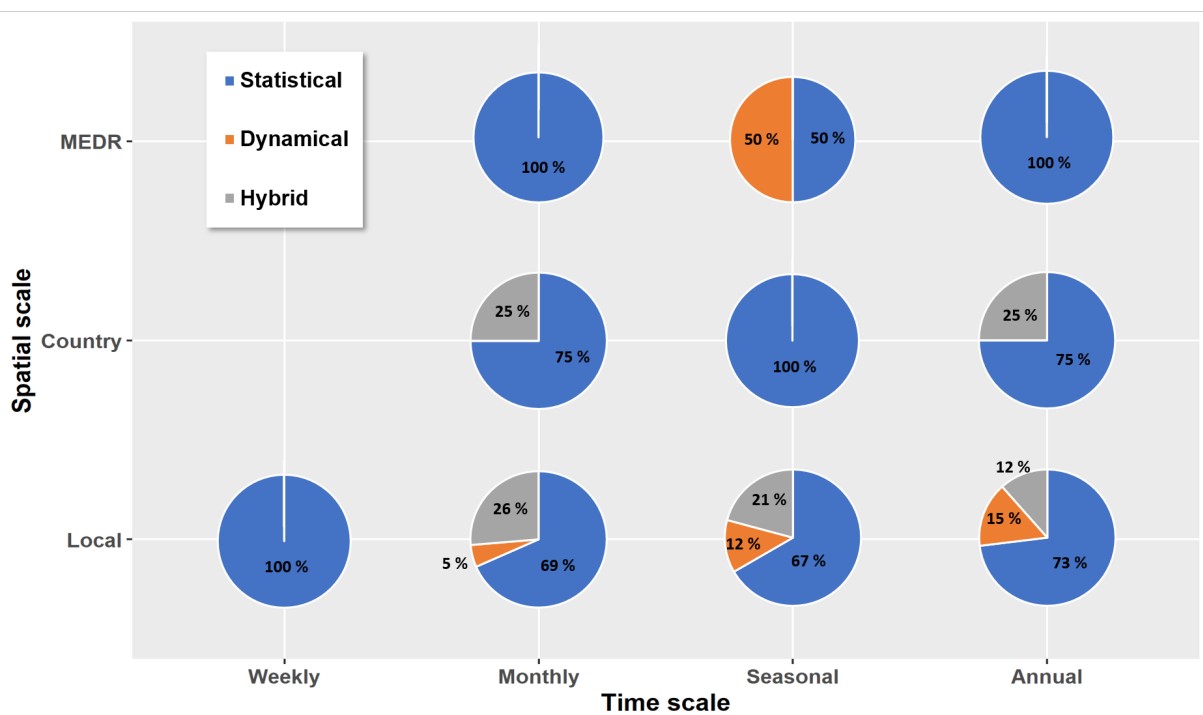


**Figure 5 Spatial and temporal scales of drought forecasting studies in the MedR with pie chart indicating the percentage**
**of use of drought forecasting method: statistical, dynamical and hybrid-statistical models for each spatio-temporal scale.**