# Peer review of "Review article: Towards Improved Drought Prediction in the Mediterranean Region – Modelling Approaches and Future Directions"

_Natural Hazards and Earth System Sciences, 2023_

## Author Comment (AC1)

**Response to the referee 1**

**MS No.: nhess-2023-63**

We sincerely thank the reviewer for the positive evaluation and valuable comments of our manuscript. We are also very grateful for your recommendations for minor revisions which we believe will greatly enhance the quality and clarity of our paper. Accordingly, the revised manuscript has been improved with new information and additional interpretations. Below we describe our responses (in normal font) point-by point to each comment (in bold text). In addition, we indicate revisions in the updated manuscript by a yellow highlighter together with the line number.

**Line 25: You define a new acronym for the Mediterranean Basin, but in the abstract, you define an acronym for the Mediterranean Region - are these the same thing? If so, just stick to one acronym. It would also be a good idea to define the region in terms of the latitude/longitude bounding box.**

Thank you for your insightful comment. Indeed, our intention was to refer to the same geographical entity, which inadvertently led to the use of both "Mediterranean Basin" and "Mediterranean Region". We recognize the confusion this may have caused and have revised our manuscript to use a consistent terminology. To avoid ambiguity, we have stuck to the term "Mediterranean Region" (MedR) throughout the manuscript.

Regarding the definition of the region, we agree that providing an approximation in terms of latitude/longitude could help give readers a clearer understanding of the geographical extent of the study. However, it is important to note that the boundaries of the Mediterranean region are not strictly defined, as they can vary depending on the specific context and criteria used. In the context of our study, we roughly considered the Mediterranean region to encompass an area between 30°N to 46°N in latitude and 5°W to 40°E in longitude. This approximation includes key countries that have a Mediterranean climate, covering the southern European coast, the northern African coast, and the eastern Mediterranean countries. Nonetheless, we acknowledge that this is just an approximation, and the precise boundaries may vary.

In response to your proposition, we have added the coordinates to the caption of figure 1 to visually represent the spatial extent of our study area. The revised caption reads now as follows:

Line 112: "Figure 1: Topography of the Mediterranean Region, approximately defined by 30°N to 46°N in latitude and 5°W to 40°E in longitude."

**Line 25: Plurality disagreement between "this kind" and "climate events" - suggested change: adaptation to this kind of climate event has been"**

Thank you for your comment. We agree with your observation about the plurality disagreement. the sentence has been revised considering your suggestion (line 29).

**Line 29: Need a comma after "For these reasons", and "Hotspot" doesn't need to be capitalized**

Thank you for your comment. We have inserted the recommended comma and corrected the capitalization of "hotspot" in the revised manuscript.

**Line 37: You should cite something about desertification and biodiversity loss**

Thank you for your comment. We agree with the reviewer's suggestion. We have now added relevant citations that discuss desertification and biodiversity loss.

Line 40-42: "This will surely lead to irreversible biodiversity loss and diminish the capability of semi-arid Mediterranean ecosystems to function as effective carbon sinks in the future ==(Valentini et al., 2000, Briassoulis, 2017; Zeng et al., 2021)==."

**Line 38: The sentence ends abruptly with "forthcoming".**

Thank you for your attentive observation. We agree that the sentence in question ends abruptly and does not fully convey our intended meaning. Here's the revised sentence: Lines 40-42: "==This will surely lead to irreversible biodiversity loss and diminish the capability of semi-arid Mediterranean ecosystems to function as effective carbon sinks in the future (Valentini et al., 2000, Briassoulis, 2017; Zeng et al., 2021)=="

**line 38: This sentence is awkward: "All these conditions exacerbate water stress that enhances in turn the probability of wildfire." Suggested change: These conditions exacerbate water stress, which, in turn, enhances the probability of wildfire."**

Thank you for your suggestion. We agree that the recommended revision significantly enhances the readability and comprehension of the sentence. As such, we have incorporated your proposed change into the revised manuscript.

Line 42: "==These conditions exacerbate water stress, which, in turn, enhances the probability of wildfire (Turco et al., 2017a)==."

**Line 84: Latter should be later**

We agree with your observation. This was an oversight on our part, and we have rectified this in the updated version of the manuscript. Your feedback is greatly valued, thank you.

**Line 94: Focus should be focuses**

Thank you. This was corrected in the revised manuscript.

**Line 197: No need for the "if"**

We agree with your comment. We have duly made this modification in the revised version of the manuscript.

**Line 198: Suggest changing "this figure tends to" to "which is expected to"**

Thank you for your suggestion. We have incorporated this revision into the updated manuscript at line 232.

**Line 482: resources shouldn't be plural**

Thank you for pointing out this error. The necessary correction has been implemented in the updated manuscript at line 521.

**Lines 602-604: Could you expand on this thought a bit? I find it to end a bit ambiguous. Could you maybe describe some of the biases that exist in some of the models that would warrant exclusion from using them in the MEDR?**

We appreciate your request for further clarification on the selection of the most skilled GCMs ensembles and their respective biases which would warrant their exclusion in the MedR. Here's a more detailed explanation:

In the context of drought prediction, selecting the most skilled GCM ensembles for the MedR involves a careful assessment of their ability to simulate the large and synoptic scale atmospheric and land-surface conditions associated with drought development. This includes evaluating their performance in terms of temperature, precipitation, pressure patterns, and other critical parameters associated with drought phenomena in the region. There are inherent biases in GCMs that could lead to their exclusion. For instance, some GCMs may have a systematic overestimation or underestimation of key meteorological variables such as temperature or precipitation. In the MedR, which is characterized by a unique mix of continental and maritime climates, some models might struggle to accurately capture the spatial and temporal variability of these climatic factors, leading to poor drought predictions. Furthermore, biases can arise in simulating the complex topography of the region, its diverse land cover and land use characteristics, and the interactions between land and atmosphere. We would like to draw attention to the choice of the MME that have demonstrated superior performance in reproducing these conditions and thus provide more accurate drought forecasts in the MedR. However, the task is challenging and necessitates rigorous validation exercises for each GCM candidate, as well as continuous monitoring and updating of selected ensembles in line with the advancements in modeling technology.

In the revised manuscript, we now have a subsection entitled Multi-Model Ensemble where we explicitly detail this explanation. Lines 669-672: "By prioritizing ensembles that adequately capture the region's distinct climate characteristics, spatial-temporal variability, and land-atmosphere interactions, the MME forecasts can mitigate biases related to key meteorological variables such as temperature or precipitation and significantly improve the precision and reliability of drought predictions (Li et al., 2023; Ahmed et al., 2019)."

**Section 5: You only mentioned one study (Bağçaci et al. 2021) that looked at CMIP6 GCMs. Are there more out there that you could review or is this a deficiency in the literature? If it is a deficiency in the literature, definitely drive that point home.**

Thank you for your valuable comment. Indeed, the citation of Bağçaci et al. (2021) was meant to illustrate one of the recent and comprehensive analyses conducted on CMIP6 GCMs. There was a growing body of literature examining CMIP6 GCMs which has not been referenced in the manuscript. This was not meant to suggest a deficiency in the literature but rather to focus on the selected study due to its comprehensiveness and relevance to the topic at hand. However, we appreciate your recommendation to clarify this point in the manuscript. We have reviewed and incorporate additional relevant studies on CMIP6 GCMs.

Lines 718-725: "In a study conducted by Cos et al. (2022), the authors compared climate projections from CMIP5 and CMIP6 models to assess the impacts of climate change in the MedR. The findings reveal a robust and significant warming trend across all seasons, with CMIP6 models projecting stronger warming compared to CMIP5. While precipitation

changes show greater uncertainties, a robust and significant decline is projected over large parts of the region during summer by the end of the century, particularly under high emission scenarios. Seker and Gumus (2022) use 22 global circulation models from CMIP6 to project future precipitation and temperature changes in the MedR. The MMEs outperform individual GCMs in simulating historical data, and the projections indicate a decrease in precipitation by 15% for SSP2–4.5 and 20% for SSP5–8.5''.

**Lines 644-645: You may want to explain what SSPs are. This is the first introduction of them and in the previous section you really only mention the RCPs from CMIP5.**

Thank you for your insightful comment. We agree that the introduction of Shared Socioeconomic Pathways (SSPs) in the text could have been more clearly explained, and we appreciate your suggestion. Indeed, adding this explanation as a footnote could serve to provide necessary information without interrupting the flow of the main text. Here is how the footnote could be added:

Page 20, line 706.

[1] SSPs are the latest climate change scenarios used in CMIP6. They not only incorporate greenhouse gas emissions scenarios like their predecessor, RCPs from CMIP5, but also integrate socioeconomic factors, such as population growth, economic development, and technological progress. Essentially, SSPs provide a more holistic view of possible future climate scenarios by considering both environmental and societal changes.

**References**

Ahmed, K., Sachindra, D.A., Shahid, S., Demirel, M.C., Chung, E.-S., 2019. Selection of multi-model ensemble of general circulation models for the simulation of precipitation and maximum and minimum temperature based on spatial assessment metrics. Hydrology and Earth System Sciences 23, 4803–4824. https://doi.org/10.5194/hess-23-4803-2019

Cos, J., Doblas-Reyes, F., Jury, M., Marcos, R., Bretonnière, P.-A., Samsó, M., 2022. The Mediterranean climate change hotspot in the CMIP5 and CMIP6 projections. Earth System Dynamics 13, 321–340. https://doi.org/10.5194/esd-13-321-2022

Li, X., Fang, G., Wei, J., Arnault, J., Laux, P., Wen, X., Kunstmann, H., 2023. Evaluation and projection of precipitation and temperature in a coastal climatic transitional zone in China based on CMIP6 GCMs. Clim Dyn. https://doi.org/10.1007/s00382-023-06781-z

Seker, M., Gumus, V., 2022. Projection of temperature and precipitation in the Mediterranean region through multi-model ensemble from CMIP6. Atmospheric Research 280, 106440. https://doi.org/10.1016/j.atmosres.2022.106440

Valentini, R., Matteucci, G., Dolman, A.J., Schulze, E.-D., Rebmann, C., Moors, E.J., Granier, A., Gross, P., Jensen, N.O., Pilegaard, K., Lindroth, A., Grelle, A., Bernhofer, C., Grünwald, T., Aubinet, M., Ceulemans, R., Kowalski, A.S., Vesala, T., Rannik, Ü., Berbigier, P., Loustau, D., Guðmundsson, J., Thorgeirsson, H., Ibrom, A., Morgenstern, K., Clement, R., Moncrieff, J., Montagnani, L., Minerbi, S., Jarvis, P.G., 2000. Respiration as the main determinant of carbon balance in European forests. Nature 404, 861–865. https://doi.org/10.1038/35009084

---

## Author Comment (AC2)

**Response to the referee 2**

MS No.: nhess-2023-63

We sincerely thank the reviewer for the thoughtful and constructive feedback regarding our manuscript. Your recognition of the potential value in our work is indeed encouraging and we appreciate the effort you have invested in the careful evaluation of our manuscript. We understand your concerns about the clarity of certain sections in the manuscript and believe that your input will enhance our manuscript's readability and effectiveness, thereby ensuring it better serves the research community and the broader public.

In response to your comments, we have revised our manuscript to include fresh perspectives, and more comprehensive interpretations that should help clarify and strengthen the points of concern. Bellow, we provide a detailed response to each of your comments (presented in bold text), paired with our corresponding actions and clarifications (in normal font). Additionally, in the updated manuscript, we have highlighted the amended text in blue and provided corresponding line numbers for easier navigation and verification.

**General comments:**

**It can be noted that the specific contribution and added value of this work is not completely clear. It is necessary to provide more details on the novelty and contributions of the present work relative to previous studies as for instance, Tramblay et al. (2020) and Hao et al. (2018).**

Thank you for your comment. We appreciate your feedback on the clarity about the distinctiveness of our work. We have taken your suggestion into consideration and revised our manuscript to further emphasize the novelty and contributions of our study. In response to specific comments throughout the review process, we have provided additional details highlighting the particularity of our work from previous studies, such as Tramblay et al. (2020) and Hao et al. (2018).

**It is critical to clearly and precisely differentiate between predictions and projections. In many cases, the terminology associated with these concepts can generate confusion by mixing them inappropriately.**

Thank you for bringing this important matter to our attention. We recognize that the inappropriate mixing of the terms projections and predictions can generate confusion and undermine the accuracy of our work. In response to this concern, we have meticulously revised our manuscript to ensure that predictions, which pertain to specific forecasts derived from available data and models, and projections, which represent potential future outcomes under specific scenarios or assumptions, are distinctly defined and consistently used throughout the document.

**It is crucial to clearly distinguish and specify the type of forecast considered, whether short-term or long-term, and to provide a precise definition of the time periods involved. This detailed specification is essential for effective communication and proper understanding.**

Thank you for your valuable comment. We completely agree with the importance of clearly specifying the spatial scale of the forecast. We have taken your suggestion into consideration and thoroughly revised our manuscript to ensure that the time scale is explicitly clarified in each referenced study and section.

**It is necessary to improve the linearity and strengthen the organization of the text to achieve greater reading fluency. It is also important to adequately support the statements made with references. I think it might be beneficial to summarize the first part of the paper (sections 1-3) to avoid redundancies.**

Thank you for your valuable feedback. We agree with your assessment regarding the organization and reading fluency of the text and have made substantial revisions in our manuscript to address these concerns.

In terms of supporting the statements made with references, we have made sure to provide appropriate and relevant citations throughout the manuscript to strengthen the credibility of our statements.

Regarding the suggestion to summarize the first part of the paper (sections 1-3), we have taken your recommendations in the specifics comments into consideration and have revised the information in these sections to avoid redundancies. This allows for a more streamlined presentation of the key points and enhances the overall flow of the manuscript.

**The figures presented in the document pose some difficulties in terms of clarity, as no information is provided on the sample used to generate the graphs, which prevents a correct interpretation of the graphs. In addition, a lack of consistency in the representation of the different prediction models is observed in Figure 4, where a separation is made for the statistical models, but the same is not done for the dynamic and hybrid models. Therefore, it is suggested to add a clear distinction between the latter methods.**

Thank you for your constructive feedback. We would like to clarify that the figures in our manuscript were constructed based on the information collected and compiled in Tables 1 through 7. To make this more evident, we have explicitly mention this in the captions of the relevant figures and in the associated text within the revised manuscript.

Figure 3 provides an overview of the drought indices used by the reviewed studies over the MedR. We have enhanced our figure caption and text to better communicate this intent.

Figure 4 presents a comparison of the accuracy of various methods based on the same evaluation criterion, which is the coefficient of determination (R2) with a 1-month lead time. We understand the need for clarity here

and have augmented our explanations in both the figure caption and the main text. In addition, we have referenced the supplementary table provided in the annex, which further details this comparison.

We also understand your point about the lack of separation in dynamical and hybrid models compared to statistical models in Figure 4. The primary reason for this is the current limited number of studies using dynamical models compared to statistical models, which has resulted in a considerably smaller number of studies available for comparison. Therefore, the separation we made for statistical models was not feasible for dynamical and hybrid models due to this limited sample size. We have elaborated on this point in the revised manuscript to avoid any potential confusion.

[Figure]

*Table 1. Studies Utilized for Comparing the accuracy of Drought Prediction Models, as Depicted in Figure 4..*

| Reference | drought approach | Forecast model | Accuracy (R²) |
|---|---|---|---|
| (Sousa et al, 2011) | Statistical | Regression | 0,79 |
| (Martínez-Fernández et al, 2016) | Statistical | Regression | 0,72 |

| | | | |
|---|---|---|---|
| (Tigkas and Tsakiris, 2015) | Statistical | Regression | 0,91 |
| (Achite et al, 2022) | Statistical | Time series | 0,97 |
| (Pablos et al, 2017) | Statistical | Time series | 0,8 |
| (Jiménez-Donaire et al, 2020) | Statistical | Time series | 0,82 |
| (Di Nunno et al, 2021) | Statistical | AI | 0,9842 |
| (El Aissaoui et al,2021) | Statistical | AI | 0,9 |
| (Achour et al, 2020) | Statistical | AI | 0,912 |
| (Habibi et al, 2018) | Statistical | Markov Chain | 0,95 |
| (Nalbantis and Tsakiris, 2009) | Statistical | Markov Chain | 0,88 |
| (Cancelliere et al, 2007) | Statistical | Markov Chain | 0,895 |
| (El Ibrahimi and Baali, 2018) | Statistical | Hybrid-statistical | 0,98 |
| (Özger et al, 2020) | Statistical | Hybrid-statistical | 0,93 |
| (Brouziyne et al, 2020) | Dynamical | Dynamical | 0,72 |
| (Mendicino et al, 2008) | Dynamical | Dynamical | 0,86 |

**It is important to clarify whether the studies mentioned are worldwide, for the Mediterranean or local. Although this distinction is made in Tables 1-7, it is not clear from reading the text.**

Acknowledging the validity of your remark, we have revised the manuscript to more explicitly state the geographical focus of each study referenced without having to refer to the tables. This should improve the flow of the narrative and make the information more accessible to the reader. Thank you for your valuable input.

**Specific comments:**

**Line 12: remove 'extremes', as these are not always extreme events.**

Thank you for your thoughtful comment. You raise a pertinent point about the changing nature of what we consider 'extreme'. In our manuscript, we refer to droughts and heatwaves as 'extreme' events based on their intensity, duration, and the significant impacts they can have. However, you're right in pointing out that these events may not always meet the statistical rarity criterion, especially as climate change is leading to their increasing frequency. In fact, in many Mediterranean regions, particularly those experiencing structural changes to their climate, these so-called 'extreme' events are starting to become the norm rather than rare deviations.

Considering your comment, we agree it would be more accurate to avoid the term 'extreme' when referring to droughts and heatwaves, to prevent any potential confusion. We have modified our manuscript accordingly.

**Lines 14-16: It would be appropriate to indicate the forecast scale, specifying whether it refers to the daily, monthly or decadal scale, as well as the type of drought considered, whether meteorological, agricultural or hydrological.**

Thank you for your valuable comment. As this paper aims to provide a holistic overview of drought prediction methodologies, we have considered all types of droughts, namely meteorological, agricultural, and hydrological, and have not confined our review to a specific forecast scale. This includes weekly, monthly, seasonal and annual forecasts. Our objective was to create a comprehensive analysis of drought forecasting models across various spatial and temporal scales to identify current knowledge gaps and opportunities for advancement. This approach allows us to understand the broad landscape of drought prediction studies, including the strengths and limitations of different models at different scales. Nevertheless, we recognize the importance of specifying this information to provide clarity to our readers. We have revised the manuscript as follows:

Lines 14-18: "This paper reviews the current state of drought modeling and prediction applied in the MedR, including statistical, dynamical, and hybrid statistical-dynamical models. By considering the multifaceted nature of droughts, the study encompasses meteorological, agricultural, and hydrological drought forms and spans a variety of forecast scales, from weekly to annual timelines. Our objective is to pinpoint the knowledge gaps in literature and propose potential research trajectories to improve the prediction of droughts in this region."

**Lines 16-18: This is not clear, only for one forecast month?**

We appreciate the reviewer's attention to detail. In our manuscript, when we mention "extended lead times," we are referring to predictions that extend beyond just the immediate future (for example, the next month). This can include several months or even seasons ahead, which is particularly beneficial for strategic planning in agriculture, water management, and other sectors that are significantly affected by drought conditions. To clarify this point, we have revised the statement as follows:

Lines 18-20: "The review finds that while each method has its unique strengths and limitations, hybrid statistical-dynamical models appear to hold the most promising potential for skillful prediction with seasonal to annual lead times."

**Lines 32-38: This paragraph needs to be revised and improved. It would be advisable to provide references to support the statements made. In addition, it is suggested to introduce in advance the different types of droughts before delving into modeling and prediction approaches.**

We are thankful for the reviewer's insightful feedback. we agree that the paragraph could benefit from more references to support the statements. In the revised manuscript, we have added multiple references to provide further substantiation of our claims regarding the projected increase in drought frequency and severity, as well as the potential for increased desertification in the Mediterranean Region due to rising temperatures.

The revised paragraph now reads:

Lines 35-39: "This report projected an increase in the frequency and/or severity of agricultural and ecological droughts across the Mediterranean and Western Africa (IPCC, 2021). A global increase of 2 °C is thought to correspond to a 3 °C increase in the daily maximum temperature in the MedR (Seneviratne et al., 2016; Vogel et al., 2021). If this increase in temperature continues at the same pace, MedR is susceptible to experience fearful desertification by the end of the 21st century, driving an increase in aridity (Carvalho et al., 2022)."

**Lines 38-39: It is recognized that a reference is missing at this point. It is suggested to read the study by Turco et al. (2017a) to get a better understanding on the mentioned topic.**

We appreciate the reviewer's suggestion to reference the study by Turco et al. (2017a) for a better understanding of the impact of changing climatic conditions on wildfire risks in the Mediterranean region. Considering this, we have reviewed the mentioned study and found it provides valuable insights that strengthen our argument. We have incorporated it into our manuscript to support our statements.

The revised sentence now reads:

Lines 40-43: "This will surely lead to irreversible biodiversity loss and diminish the capability of semi-arid Mediterranean ecosystems to function as effective carbon sinks in the future (Valentini et al., 2000, Briassoulis, 2017; Zeng et al., 2021) . These conditions exacerbate water stress, which, in turn, enhances the probability of wildfire (Turco et al., 2017a)"

**Line 51: It is recommended to add 'generally', as this is not always the case: 'with generally decreasing precipitation values towards the south'.**

Thank you for your suggestion. We have revised the sentence in the manuscript to include "generally" as you recommended. (See line 54)

**Line 58: it is recommended to add a reference to this statement.**

We agree with reviewer's suggestion. We have now added relevant citation that discusses the structural water shortages in these regions and their expected exacerbation with future population growth.

Line 59-62: "Accordingly, several countries such as Algeria, Morocco, Egypt, Libya, Malta, and some countries of southern Europe such as Portugal and Spain are experiencing a structural water shortage that is likely to increase with the expected population growth (Sanchis-Ibor et al., 2020)."

**Line 60: It is not clear how far in advance is considered in this statement. It would be helpful to provide additional information on the lead time used.**

Thank you for your comment. We understand the need for clarity on the term "sufficient lead time." This can indeed vary based on the context and specific needs of water resource management and agriculture.

In the context of our manuscript, "sufficient lead time" generally refers to a forecasting window that allows for effective proactive measures to be put in place. This can range from a few weeks to several months, depending on the type of drought being forecasted (meteorological, agricultural, or hydrological) and the specific mitigation strategies. For instance, meteorological droughts might require shorter lead times of a few weeks to a couple of months, while agricultural and hydrological droughts, due to their slower onset, could benefit from forecasts made several months in advance.

The revised sentence now reads:

Lines 62-64: "In this challenging context, drought forecasting that provides seasonal to annual lead times becomes critically important for proactive agricultural and water resources management."

**Lines 64-67: I suggest to mention also climatology-based or persistence-based statistical models, as they represent possible baselines for drought prediction. For example, I suggest to include also the Ensemble streamflow prediction (ESP) system (Day, 1985; Twedt et al., 1977; AghaKouchak, 2014a; Turco et al., 2017b; Torres-Vázquez et al., 2023).**

Thank you for your comment. We have incorporated your suggestion as follows:

Lines 68-73: " Statistical models, also known as data-driven models, rely on the estimated correlations between several predictors (large-scale climate variables) and predictands (local climate variables represented by historical observations). The climatology-based or persistence-based models, like the Ensemble Streamflow Prediction (ESP) system, form an essential tool in this category, leveraging both historical and near real-time data to generate a probabilistic forecast of future drought events (AghaKouchak, 2014a; Turco et al., 2017b; Torres-Vázquez et al., 2023."

**Line 80: I suggest to restructure this section together with lines 50-54, where a detailed description of the Mediterranean region is given.**

We appreciate your suggestion to restructure the section for better flow and continuity in the description of the Mediterranean region. We have revised the manuscript accordingly.

Lines 52-53: Notably, the MedR is positioned in a transitional band between the midlatitude and subtropical regions, which makes climate modeling for this region quite challenging (Planton et al., 2012).

**Lines 91-95: It is necessary to provide more details on the novelty and contributions of the present work in relation to the previous study by Tramblay et al. (2020) and Hao et al. (2018).**

Thank you for raising this important point. We acknowledge the significant contributions of both Tramblay et al. (2020) and Hao et al. (2018) to the field of drought prediction, and our study seeks to build upon their foundational work, bringing a new and important perspective to this area of research.

While Hao et al. (2018) provided a comprehensive overview of drought prediction at a global scale, our paper offers an in-depth analysis of drought prediction methodologies specifically applied to the Mediterranean context. This is achieved through an examination of the applicability, strengths, and limitations of statistical, dynamical, and hybrid statistical-dynamical models, in line with the regional specifics of the Mediterranean region. This specificity is vital given that drought, as a phenomenon, is highly region-dependent. The unique meteorological conditions of the Mediterranean region necessitate dedicated studies, as solutions developed for other regions may not be applicable or effective here.

Tramblay et al. (2020) focused on the challenges related to drought assessment in the Mediterranean region under future climate change scenarios. They did underscore the need for drought modelling and forecasting tailored to the Mediterranean context as climate change continue to exacerbate drought conditions in this region. Our work builds on this by not just emphasizing the complexity of drought assessment but also reviewing recent drought forecasting methods applied to the region. Our investigation not only sheds light on the merits and limitations of these methods but also helps identify underexplored areas that warrant further research. Detecting these gaps is a crucial aspect of our work, helping direct future research towards these relatively unexplored realms of drought prediction. As such, our study offers a guide for future efforts aimed at improving our understanding of drought prediction in the Mediterranean region and potentially other areas.

We have revised our manuscript to make these unique contributions more apparent and believe that these modifications more clearly illustrate how our study complements and extends the works of Tramblay et al. (2020) and Hao et al. (2018).

Lines 90-106: "To address these questions, numerous review papers have sought to consolidate the scientific advances in drought prediction from different regions of the world (e.g., Mishra and Singh, 2011; Hao et al., 2018; Fung et al., 2019; Han and Singh, 2020). While these studies provided a comprehensive overview of drought prediction at a global scale, our paper offers an in-depth analysis of drought prediction methodologies specifically applied to the Mediterranean context. This is achieved through an examination of the applicability, strengths, and limitations of statistical, dynamical, and hybrid statistical-dynamical models, in line with the

regional specifics of the MedR. This specificity is vital given that drought, as a phenomenon, is highly region dependent. The unique meteorological conditions of the MedR necessitate dedicated studies, as solutions developed for other regions may not be applicable or effective here.

Tramblay et al. (2020) emphasized the urgent need for drought modeling and forecasting methods designed for the Mediterranean context, particularly as climate change continues to exacerbate drought conditions in this region. Building on this, our work not only emphasizes the complexities of drought assessment but also conducts a critical review of recent drought forecasting methodologies applied specifically to the MedR. In addition to shedding light on the merits and limitations of these methods, our investigation also helps identify underexplored areas that warrant further research. Detecting these gaps is a crucial aspect of our work, as it directs future research towards these relatively unexplored realms of drought prediction."

**Lines 106-107: Recommend adding a reference to this statement to adequately support it.**

Thank you for your comment. We have added a reference that support our statement.
Line 116: "The multidisciplinary and multiscale nature of drought renders the understanding of this phenomenon very challenging (AghaKouchak et al., 2021)."

**Lines 112-113: Change the citation structure to fit the appropriate form. For example, "see for example Crausbay et al. (2017) ..."**

Thank you for your comment and guidance. We have revised the citation structure as you suggested (Lines 122-123).

**Line 120: It is recommended to add a reference to this statement to adequately support it.**

Thank you for your comment. We have added a reference that support our statement.
Line 129-131: "However, the lack of a universal definition of drought is also apparent in the huge variety of indices (more than 100) that have been developed for drought prediction (Lloyd-Hughes, 2014)."

**Lines 124-136: It is recommended to refer to the definitions provided by WMO and IPCC to get a more precise understanding of the terms used in the context of droughts and climate change.**

Thank you for your insightful comment. We agree that referring to the authoritative definitions provided by the WMO and IPCC can help to provide a more precise and universally accepted understanding of drought. We have revised this paragraph as follows:

Lines 134-141: "The World Meteorological Organization (WMO) characterizes meteorological drought as "a prolonged absence or marked deficiency of precipitation". Similarly, the IPCC defines meteorological drought as "a period of abnormally dry weather in a region over an extended period". The threshold to distinguish between a dry or wet period often depends on the average rainfall typical for the specific area under study. This gives rise to a variety of meteorological definitions, each tailored to the distinct conditions of diverse regions or

countries (Isendahl, 2006). Regarding the MedR, creating a single encompassing definition of meteorological drought is particularly challenging. This complexity stems from the diverse climate conditions across the region, particularly the pronounced variability between eastern and western meteorological conditions that contribute to drought."

**Line 139: It is not necessary to use the word 'onset'. It is recommended to remove it for clarity and conciseness.**

We agree with your recommendation and have revised the sentence accordingly.

**Line 155: It should indicate what it correlates with to provide a more accurate understanding.**

Thank you for your comment. We agree that providing explicit correlation information would indeed improve understanding. Russo et al. (2019) performed drought characterization in the Mediterranean region using both SPEI and SPI, considering the period from 1980 to 2014. Their study aimed to identify whether the occurrence of extremely hot days/nights in summer was preceded by drought events in spring and early summer. They achieved this by correlating the number of hot days and nights in the hottest months with a drought indicator from the preceding months. Findings from their study indicated that SPEI exhibits a stronger correlation with early drought conditions over a 3-month timescale, while SPI showed a better correlation with drought over a 9-month duration. This result suggests the superior capability of the SPEI in capturing early shifts between evapotranspiration and precipitation, potentially providing an early warning for the occurrence of hot days/nights. Therefore, this work underscores the value of using SPEI for drought characterization and forecasting in the Mediterranean region (Russo et al., 2019).

We have revised line 155 as follows:

Lines 171-175: "

Russo et al. (2019) performed drought characterization in MedR using both SPEI and SPI, considering the period 1980–2014. Their findings indicated that SPEI exhibits a stronger correlation with drought conditions over a 3-month time scale, while SPI shows a better correlation for a 9-month duration. This result highlights the ability of SPEI to capture the early shifts in the balance between evapotranspiration and precipitation more efficiently than SPI (Russo et al., 2019)."

**Line 157: The statement is not clear. It is recommended to rewrite or provide more details to improve the reader's understanding.**

Thank you for pointing out the ambiguity in this statement. We agree that further explanation is needed. Here is a revised version:

Lines 176-181: "Despite the utility of SPEI in drought characterization, it does have a noteworthy limitation. The effectiveness of SPEI significantly relies on the method used for estimating PET such as the Penman-Monteith equation, the Thornthwaite method, the Hargreaves method, and the Priestley-Taylor method. These estimation methods can yield varying results, leading to inconsistencies in SPEI values. In essence, the sensitivity of SPEI to the PET estimation method used could potentially affect the accuracy and reliability of the index in representing drought conditions (Vicente-Serrano et al., 2010; Stagge et al., 2014)."

**Line 167: It is recommended to consult the study by Vicente-Serrano et al. (2012).**

Thank you for your recommendation. We have duly incorporated this information into our manuscript. Specifically, we have included the conclusion from Vicente-Serrano et al. (2012) that SPEI, which factors in both precipitation and evapotranspiration, exhibited superior performance in capturing drought impacts, especially during the critical summer season. This point has been added to reinforce and complement the statement made by Ionita and Nagavciuc (2021) regarding the limitations of the SPI in fully revealing drought variability.

Lines 166-167: "A global assessment of drought indices conducted by Vicente-Serrano et al. (2012) found that SPEI provided a superior capability in capturing drought impacts, particularly during the crucial summer season."

Lines 191-193: "This underscores the findings of Vicente-Serrano et al. (2012), who emphasized the benefits of using more integrative indices like SPEI in understanding and predicting drought variability more effectively."

**Line 170: Change the citation structure to fit the proper form. For example: "Paulo. (2012) compared ..."**

Thank you for your comment and guidance. We have revised the sentence as you suggested (line 196).

**Line 177: It is recommended to add a reference to support this statement. Also, it is important to keep in mind that in many Mediterranean regions these crops may be controlled by irrigation systems.**

We agree with your comment regarding the presence of both rain-fed and irrigated agriculture systems in the Mediterranean basin. We have modified our text accordingly.

The revised text reads: Line 202-205: "In the Mediterranean basin, agricultural practices span both rain-fed and irrigated systems. Rain-fed agriculture is prevalent, particularly for crops such as wheat and barley, while crops like olives and citrus fruits, such as oranges, often utilize controlled irrigation systems to supplement natural precipitation (Rodrigo-Comino et al., 2021)."

**Lines 178-179: It is recommended to add a reference to support this statement.**

Thank you for your comment. We have added a reference that support our statement.

Line 205-208: "Regardless of the system employed, if meteorological drought lasts for a prolonged period, it can lead to a reduction in soil moisture to such a level that it harmfully affects crop production, especially during the active plant growth season (Wilhite and Glantz, 1985; Mishra and Singh, 2010). At this stage the agricultural drought sets in."

**Lines 188-195: This statement is not correct. There are other techniques and methods to obtain soil moisture. It is recommended to read about this topic. For instance, it is recommended Tramblay et al. (2019), Miralles et al. (2010) and Martens et al. (2017).**

Thank you for your valuable input and the recommended references. We acknowledge that there are indeed other methods to measure soil moisture besides remotely sensed data, such as in-situ soil moisture probes and cosmic-ray neutron probes. Our intention was to underscore the advantage of remotely sensed data in terms of spatial coverage and frequency of measurements. However, we agree that our phrasing may have unintentionally conveyed the notion that it is the only method available. To rectify this, we have revisited the pertinent literature, including Tramblay et al. (2019), Miralles et al. (2010), and Martens et al. (2017), and updated the manuscript accordingly.

The revised statement reads:

Lines 216-225: "The formulation of these indices integrates soil moisture data, leveraging a variety of assessment techniques, each with unique advantages. These include in-situ soil moisture probes, cosmic-ray neutron probes, and physically driven models such as the ISBA land surface model (Tramblay et al., 2019). Each of these techniques has distinct advantages and is suitable for different application contexts (Miralles et al., 2010; Martens et al., 2017). However, when faced with the scarcity of observed soil moisture data, remote sensing comes to the forefront. It furnishes extensive and frequent measurements of soil moisture characteristics, effectively supplementing areas where observed data falls short. Yet, it is crucial to be aware of the limitations of these tools. Despite its indispensable role, remote sensing is constrained by factors such as coarse temporal and spatial resolution, limited penetration depth, and incompatible governing hydrologic principles (Mohanty et al., 2017; Gruber and Peng, 2022). "

**Line 196: It is recommended to add a reference to support this statement.**

Thank you for your feedback. We agree that this statement would be strengthened by a supporting reference. To this end, we have added a reference, which is as follows:

Lines 230-231: "In general, when soil moisture in the root zone reaches a critical level, farmers resort to irrigation to save crops (Kang et al., 2002)"

**Line 223: Change the citation structure to fit the appropriate form.**

Thank you for your comment and guidance. We have revised the citation form.

**Lines 234-236: Unclear, aspects such as whether it refers to SPI-12 or SDI-6 should be clarified.**

Thank you for your valuable feedback. In response to your comment, we have clarified this statement as follows:

Lines 267-273: Bouabdelli et al. (2020) conducted a comparison study of the SPI and SDI, focusing on their characteristics across three watersheds in northwestern Algeria. Their analysis revealed a substantial similarity between meteorological drought events (as represented by SPI-12) and hydrological drought events (as indicated by SDI-6). This correlation emphasizes the sensitive and responsive nature of these basins to dry conditions, further illustrated by the swift transition from meteorological to hydrological drought events in the studied basins (Bouabdelli et al., 2020).

**Line 239: It is recommended to add a reference to support this statement.**

We agree that adding a reference would support our statement. Here's the revised sentence with a reference included:

Line 274-276: "The application of hydrological drought indices appears to be very valuable. However, the main challenge in applying these indices lies in the requirement for a long-term series of climatic data. According to WMO, up to 30 years of continuous rainfall data may be necessary for accurate drought index calculations (WMO, 1994)."

**Lines 250-251: This is not clear. The forecast time being referenced should be specified. For example, if seasonal forecasts are involved, they may be more reliable months in advance, whereas daily forecasts may have limitations from two weeks. We suggest reading the study by Lorenz (1963) for more information on this topic.**

Thank you for your comment. We agree that the time scale of the forecast should be specified for clarity. In this paragraph, we were referring to the limitations of precipitation predictability, specifically in the context of meteorological drought prediction. This statement refers to the generally recognized forecast horizon of one month due to the chaotic nature of atmospheric circulation.

We understand your suggestion to further elaborate on the predictability of other drought types, like seasonal and daily forecasts, and their potential limitations. We have read Lorenz's work (1963) as you suggested and appreciate the relevance it brings to this context. We have revised our text to clarify that the "one-month lead time" is associated with meteorological droughts and the forecast of chaotic atmospheric elements that contribute to their onset. We've also added that these limitations can vary when considering different types of droughts and different forecast scales.

The paragraph now reads:

Lines 284-293: "Considering the various forms of drought, meteorological droughts, characterized by a deficit in precipitation, are commonly recognized as marking the onset of drought conditions. This initial stage is intrinsically linked to precipitation predictability, which is driven by large-scale atmospheric motions such as Walker circulations and Rossby waves, influenced by factors like SST anomalies, radiative forcing changes (both natural and anthropogenic), and land surface interactions (Hao et al., 2018; Wood et al., 2015). However, due to the inherently chaotic nature of atmospheric circulation, predictability, particularly for meteorological droughts, tends to diminish beyond a one-month lead time. It is crucial to note that the reliability of these predictions can differ when considering other drought types (such as agricultural or hydrological droughts) or altering the forecast scale, with seasonal forecasts often displaying more reliability months in advance, while daily forecasts may face limitations from around two weeks."

**Lines 255-256: Where is it possible to accurately predict seasonal drought with a long lead time (>1 month)? It would be beneficial to provide additional details and references to support this statement.**

Thank you for your insightful comment. In our statement, we referred to the ability to predict seasonal droughts with long lead times (>1 month), particularly in areas where drought onset is strongly influenced by known ocean-atmospheric teleconnections such as ENSO. This is primarily applicable in regions within the tropics where ENSO impacts are substantial. For instance, during the peak phase of El Niño or La Niña in the tropical Pacific, a corresponding change in precipitation patterns can be observed several months later in North American winter climate (Livezey and Smith, 1999; Hoerling and Kumar, 2003). This delayed impact provides a crucial window for predicting potential drought conditions with a long lead time exceeding one month (Johnson and Xie, 2010). Moreover, this lagged correlation allows for proactive drought management strategies, with the ability to anticipate and prepare for drought conditions based on forecasted ENSO conditions.

However, we acknowledge that the predictability of drought, especially at long lead times, remains a complex issue and can vary significantly depending on the region and specific local conditions. We have revised our manuscript to better reflect this complexity and provide additional references to support our statement.

Lines 295-306: "Notably, it is widely established within the scientific community that certain ocean-atmospheric teleconnections, such as ENSO, can profoundly influence the onset of drought conditions in various regions worldwide, particularly in the tropics (Ropelewski and Halpert, 1987; Shabbar and Skinner, 2004; Hoell et al., 2014; Vicente-Serrano et al., 2017). For instance, during the peak phase of El Niño or La Niña in the tropical Pacific, a corresponding change in precipitation patterns can be observed several months later in North American winter climate (Livezey and Smith, 1999; Hoerling and Kumar, 2003). This delayed impact provides a crucial window for predicting potential drought conditions with a long lead time exceeding one month (Johnson and Xie, 2010). Moreover, this lagged correlation allows for proactive drought management strategies, with the ability to anticipate and prepare for drought conditions based on forecasted ENSO conditions.

Nevertheless, drought predictability is seasonally and spatially variable. Typically, the accuracy of seasonal drought prediction is superior in the tropics, while it stills challenging in the extra-tropics (Doblas-Reyes et al., 2013)."

**Line 258: This statement corresponds to Doblas-Reyes et al. (2013), not the above reference.**

Thank you for pointing out this discrepancy. We appreciate your careful review and attention to detail. Indeed, you are correct, this statement should be attributed to Doblas-Reyes et al. (2013). We have corrected this citation error in the revised manuscript.

**Line 266: Can the NAO be predicted months in advance?**

Thank you for your question. The predictability of the NAO several months in advance is indeed a topic of ongoing research. Currently, the consensus within the scientific community is that the skill in predicting the NAO is generally modest, particularly beyond a seasonal time frame (Czaja and Frankignoul, 1999; Saunders and Qian, 2002). However, some studies suggest that there might be potential for predictability related to slow components of the climate system, such as sea surface temperatures, sea ice and soil moisture (e.g., Scaife et al., 2014; Dunstone et al., 2016 ). We have clarified this aspect in the revised manuscript to reflect the current state of knowledge regarding the predictability of the NAO.

Lines 311-316: "In contrast, the NAO is commonly identified as a prominent factor influencing Mediterranean climate variability during the winter season (Ulbrich and Christoph, 1999; Vicente-Serrano et al., 2011; Kahya, 2011; Santos et al., 2014; Cook et al., 2016). It is important to note, however, that while acknowledging the profound impact of the NAO on the climate dynamics of the MedR, its predictability, especially on seasonal scales, continues to be a considerable challenge in the field of climate science (Czaja and Frankignoul, 1999; Saunders and Qian, 2002; Scaife et al., 2014; Dunstone et al., 2016)."

**Lines 269-270: It is not clear, it is recommended to be more precise.**

We appreciate the referee's request for further clarification on this statement. In the revised manuscript, we provide more detail as follows:

Lines 320-325: "This analysis suggests Mediterranean droughts from 850-1849 CE were mainly driven by the internal variability of the climate system, including elements like barotropic high-pressure systems, positive NAO phases, and La Niña-like conditions. Conversely, external forcings such as volcanic eruptions were found to be associated with wetter Mediterranean conditions. In the period 1850-2099 CE, however, anthropogenic influences amplified land-atmosphere feedbacks, leading to persistent dry conditions in the Mediterranean (Kim and Raible, 2021)."

**Line 279: It is suggested to use common indices that apply even in the Mediterranean, see, e.g. https://www.cpc.ncep.noaa.gov/**

Thank you for your suggestion. We understand your point on the importance of universally accepted indices for broader applicability and comparison. In our study, we have used some of the common indices such as the SPI and the SPEI, both of which are recognized by global meteorological and climatological communities. However, the primary reason we mentioned ULMOi was to highlight the development of region-specific indices tailored to capture the unique climatic characteristics of certain areas. We believe that while common indices offer robust and standardized ways to compare across different regions, the inclusion of more tailored indices may help capture the unique climatic characteristics of the Mediterranean region.

**Lines 282-283: Change the citation structure to fit the proper form.**

Thank you for your comment and guidance. We have revised the citation form.

**Lines 284-291: Suggest removing these recurring paragraphs to avoid redundancies in the article.**

Thank you for pointing out this redundancy. We agree with your feedback and have consequently removed the repetitive sections to ensure the clarity and conciseness of our manuscript.

**Lines 292-310: This should be structured together with lines 50-54 for better organization and understanding. It would be beneficial to review whether all the information provided on the physical characteristics of the Mediterranean region is relevant and consider deleting sections that do not add substantial value. For example, lines 307-310 may be reconsidered in terms of their relevance to the central theme of the article.**

Thank you for your insightful comment. We have reviewed two paragraphs in the introduction to ensure a more coherent description of the physical characteristics of the MedR.

Lines 46-57 :" The Mediterranean Sea (MEDS), lying between Africa, Europe, and Asia, serves as a substantial source of moisture and heat, affecting atmospheric circulation and weather patterns (Mariotti et al., 2008). Its narrow connection to the Atlantic Ocean via the 14 km wide Strait of Gibraltar and the surrounding varied topography (Fig. 1), with vegetated areas to the north and desert areas to the south and east, contribute to the region's complex climate dynamics (Michaelides et al., 2018).

The MedR is characterized by a mid-latitude temperate climate with mild rainy winters and hot, dry summers (Lionello et al., 2023). Notably, this area is positioned in a transitional band between the midlatitude and subtropical regions, which makes climate modeling for this region quite challenging (Planton et al., 2012). The Mediterranean climate exhibits a strong spatial gradient in precipitation, with generally decreasing precipitation values towards the south and hardly any precipitation during the summer (Lionello, 2012). Such conditions pose

challenges in climate modeling and can lead to severe impacts on water supply and agriculture, especially in regions relying on rain-fed agriculture (Tramblay et al., 2020).")

We acknowledge your concern about the relevancy of certain information, particularly on lines 307-310. Upon reflection, we realize these details might not significantly contribute to the primary focus of the study, and thus, we have considered their removal to make the text more concise and focused.

**Line 328: The time scale used for the evaluation of model performance needs to be specified. For example, is it evaluated weekly, monthly, or on a decadal scale?**

Thank you for your comment. We agree that the time scale used for the evaluation of model performance in the reviewed studies need to be elucidated in this section.

We have modified this section to include this information.

Lines 385–390: "Bouznad et al. (2021) conducted a comparative analysis of ARIMA and SARIMA models using precipitation, temperature, and evapotranspiration data to assess seasonal drought conditions in the Algerian highlands. These models were compared based on their ability to replicate and forecast the data series accurately. The SARIMA model emerged as the better choice as it exhibited significant p-values for all variables under study. This implies that the model was statistically significant in predicting the variables and thus outperformed the ARIMA model in this specific context."

Lines 390-392: "Achite et al. (2022) investigated the meteorological and hydrological drought in the Wadi Ouahrane basin using ARIMA and SARIMA models applied to SPI and SRI indices. A validation based on $R^2$ revealed high accuracy for SPI and SRI of 0.96 and 0.97 respectively, at 1-month lag."

**Line 336: It is recommended to add a reference to adequately support this statement.**

Thank you for your recommendation. We agree that this statement would be strengthened by a supporting reference. To this end, we have added the following reference:

Line 365: "ARIMA is the most frequently used time-series model (Zhang et al., 2003)."

**Line 356: Change the citation structure to fit the appropriate form.**

Thank you for your comment and guidance. We have revised the citation form.

**Line 358: better model to reproduce a forecast series?**

Thank you for your query. Yes, the statement in line 358 indeed refers to the SARIMA model's superior performance in reproducing and predicting the time series data in the study conducted by Bouznad et al., 2021.

This study used both ARIMA and SARIMA models to analyze precipitation, temperature, and evapotranspiration data in order to assess drought conditions in the Algerian highlands. These models were compared based on their ability to replicate and forecast the data series accurately. The SARIMA model emerged as the better choice as it exhibited significant p-values for all variables under study. This implies that the model was statistically significant in predicting the variables and thus outperformed the ARIMA model in this specific context.

Lines 385–390: "Bouznad et al. (2021) conducted a comparative analysis of ARIMA and SARIMA models using precipitation, temperature, and evapotranspiration data to assess seasonal drought conditions in the Algerian highlands. These models were compared based on their ability to replicate and forecast the data series accurately. The SARIMA model emerged as the better choice as it exhibited significant p-values for all variables under study. This implies that the model was statistically significant in predicting the variables and thus outperformed the ARIMA model in this specific context."

**Line 361: It is not clear what R2 is validated with.**

A validation based on $R^2$ revealed high accuracy for SPI and SRI of 0.96 and 0.97 respectively, at 1-month lag.

**Line 389: Change the citation structure to fit the appropriate form.**

Thank you for your comment and guidance. We have revised the citation form.

**Lines 391-392: There is perceived to be missing information or a sense of incompleteness in this section. It would be advisable to revise and add additional details to provide a more complete explanation or clarify any aspects that may have been left out.**

Thank you for your suggestion. We have reviewed Sousa et al. (2011) and agree that the summary could be enhanced with additional details. Here is a revised version of the sentence:

Lines 421-423: "Their study successfully developed a robust stepwise regression model capable of predicting summer drought conditions six months in advance with a high correlation of 0.79 between simulated and observed scPDSI time series, thus demonstrating its utility in forecasting future drought conditions in the region."

**Line 402: It is necessary to specify for which waiting times are being referred to in this statement.**

Thank you for your comment. In the context of Martínez-Fernández et al., (2016)'s work, the waiting times refer to the weekly intervals at which the soil moisture data was collected and analyzed to monitor agricultural drought. These "waiting times" align with common agricultural practices as farmers frequently operate on a weekly schedule for irrigation (Purcell et al., 2003), which further underscores the practical relevance of this data collection interval for real-world applications in agricultural drought monitoring.

We have revised this statement to clarify this:

Lines 428-435: "Martínez-Fernández et al. (2016) conducted a study in the REMEDHUS (Soil Moisture Measurement Stations Network) area in Spain, aiming to monitor agricultural drought on a weekly time scale and provide early warning to farmers for adapting irrigation strategies. They computed a specific agricultural drought index (SWDI) using data from the SMOS satellite. Within this study, various computation approaches were analyzed, and the ones that yielded the most promising results were those directly based on soil attributes or parameters extracted from pedo-transfer function (PTF). These approaches utilized a multiple regression analysis, with soil water parameters as dependent variables and incorporated other relevant soil characteristics such as texture, bulk density, and porosity."

**Line 410: It is recommended to add a reference to adequately support this statement.**

Thank you for your recommendation. We agree that this statement would be strengthened by a supporting reference. We have then added the following reference:

Lines 443-444: "One of the big challenges in drought prediction is the random and nonlinear nature of the hydroclimatic variables (Agana and Homaifar, 2017)."

**Line 414: It would be interesting to clearly define the difference between 'forecasting' and 'modeling' to provide an accurate understanding of these terms and how they apply in the context of the article.**

Thank you for your insightful comment. We agree that clarifying the difference between 'forecasting' and 'modeling' would provide a better understanding of these terms in the context of the article.

Modeling in the context of drought refers to the process of creating a mathematical representation of the drought system, which includes the various factors and variables that influence the occurrence and severity of drought. Models help us understand the complex relationships between these variables and simulate different drought scenarios. On the other hand, forecasting refers to the application of these models to predict future drought conditions based on current and historical data. Forecasting uses the established models to anticipate future outcomes, allowing us to prepare and implement strategies for drought mitigation.

Therefore, our revised statement would be:

Lines 447-451: "These algorithms have thus garnered significant interest in the realms of drought modeling and forecasting (Prodhan et al., 2022). In the context of modeling, they are used to develop mathematical representations of complex drought systems, capturing the interplay of various atmospheric, hydrological, and land surface processes that lead to these phenomena. In forecasting, the models derived from these algorithms are employed to anticipate future drought conditions, assisting in risk assessment and mitigation strategies."

**Line 415: Do you mean 'intelligent'? It would be helpful to clarify this term and provide more detail on its meaning and how it applies in the context of drought forecasting.**

The term 'intelligent' in this context refers to models that leverage techniques from the field of artificial intelligence and machine learning. We have replaced the term 'intelligent models' with 'machine learning models' to avoid confusion.

The revised sentence: Lines 451-452 "Table 3 highlights key studies that utilize machine learning models for drought prediction in the MedR."

**Line 416: Change the citation structure to fit the appropriate form.**

Thank you for your guidance. We have revised the citation form.

**Line 419: It is important to clarify whether this statement refers to a global scale or specifically to the Mediterranean area.**

Prodhan et al. (2022) reviewed machine learning methods for drought monitoring and forecasting on global scale, not specifically focusing on the Mediterranean area. They observed that the Artificial Neural Network model was the most popular among the studies they reviewed, which encompassed various geographical regions worldwide.

To avoid confusion, we have clarified this in line 454 to indicate that Prodhan et al.'s (2022) findings apply to a global scale.

**Lines 435-439: It is recommended to be consistent in the number of decimal places used. I suggest using 2 decimal places.**

Thank you for your comment. We completely agree with your suggestion. In the revised manuscript, we have ensured to standardize the decimal places across all values, and we have now limited them to two decimal places as suggested.

**Line 440: SVM is the same as SVMs?**

In the text, both 'SVM' and 'SVMs' refer to Support Vector Machines. 'SVM' is the abbreviation of the term, while 'SVMs' is the plural form of the abbreviation. In the revised manuscript, we have stuck to the term "SVM" to avoid any potential confusion.

**Line 447-455: It is important to clarify whether this section is talking about modeling or forecasting.**

Thank you for your comment. This section indeed discusses both aspects: modeling and forecasting of drought events.

El Aissaoui et al. (2021) developed a SVR model, which is the modeling aspect. The model, equipped with different kernel functions (linear, sigmoid, polynomial, and RBF), was utilized to predict, or forecast, drought in the region of Upper Moulouya, Morocco. Their research underscores the SVR model's effectiveness, particularly with the RBF kernel function, in forecasting drought indices (SPI and SPEI). Similarly, Mohammed et al., (2022) examined the applicability of four different Machine Learning algorithms namely BG, RSS, RT, and RF in forecasting agricultural and hydrological drought events in the eastern MedR based on SPI. The process of evaluating these algorithms' performance can be seen as the modeling phase, and their application to predict drought conditions represents the forecasting aspect.

We have clarified this in lines 462-468 to ensure the distinction between modeling and forecasting is apparent.

Lines 480-485: "In the context of drought studies, SVM is particularly beneficial due to its ability to handle many inputs, use a small dataset for training, and its resistance to overfitting compared to ANN (Hao et al., 2018). These features make SVM less sensitive to data sample size, enhancing the robustness of the drought model. On the forecasting aspect, SVM employs a kernel function to map predictors in a high dimensional hidden space, subsequently transforming the predictand to the output space (El Aissaoui et al., 2021). This process allows the SVM model to generate effective and accurate forecasts about potential future drought events, given the input variables."

**Line 468: Change the citation structure to fit the proper form.**

Thank you for your comment and guidance. We have revised the citation form.

**Line 492: It would be useful to review AghaKouchak et al. (2014b) for more information on this topic and to support the claim.**

We appreciate your suggestion to incorporate insights from AghaKouchak et al. (2014b). This source indeed offers a deeper perspective on the utilization of copula models in the realm of hydrologic research. Accordingly, we have included this reference in our revised manuscript to support our claim (line 531).

**Line 495: Reference is missing.**

We agree that adding a reference would support our statement. Here's the revised sentence with references included:

Lines 535: The choice of the most suitable copula family depends on the specific modeling goals and the structure of the data being modeled (Genest and Favre, 2007; Joe, 2014).

**Line 500: It is important to clarify whether the studies mentioned are on a global scale or specific to the Mediterranean.**

We appreciate your observation. When referencing Aas and Berg (2009) and Savu and Trede (2010), our aim is to provide methodological insight for researchers looking to construct high-dimensional copulas. These references offer broad methodological principles, not restricted to any geographic area. Our manuscript structure usually begins with an exposition of the general principles of each method, which is then followed by a review of specific applications of this method within the Mediterranean region. This ensures our readers hold both the universal concepts and their specific applications.

We have adjusted the statement to underscore that the cited references provide foundational methodological understanding for working with copulas in high dimension, transcending specific geographical contexts.

Lines 538-540: Furthermore, comprehensive methodological understanding of constructing high-dimensional copulas, such as Pair Copula Construction (PCC) and Nested Archimedean Construction (NAC), can be garnered from the works of Aas and Berg (2009) and Savu and Trede (2010).

**Lines 531-533: There is some recurrence in this section. It would be advisable to review and eliminate unnecessary repetitions to improve the flow and clarity.**

Thank you for pointing out this redundancy. We have removed the mentioned section.

**Lines 562-571: It is not clear whether this section is talking about climate change scenarios or seasonal forecasts.**

Thank you for your comment. We have revised this section to clarify that the focus is on seasonal forecasts, rather than climate change scenarios.

Lines 624-636: "Dynamical drought prediction methods are generally based on the use of seasonal climate forecasts derived from comprehensive GCMs. The European Centre for Medium-Range Weather Forecasts (ECMWF)'s System 4 (SYS4), the Hadley Centre's Global Environmental Model (HadGEM), the Community Earth System Model (CESM), and the National Centers for Environmental Prediction (NCEP)'s Climate Forecast System (CFS) are some widely recognized examples. Designed to emulate physical processes across the atmosphere, ocean, and land surface, these GCMs can produce near-term forecasts for various climatic factors such as precipitation, temperature, surface pressure, and winds. However, these models typically provide a global overview and possess a relatively coarse resolution, which spans from 150 to 300 km horizontally, encompassing 10 to 20 vertical atmospheric layers and up to 30 oceanic layers. This level of detail may not offer the specificity necessary for local-scale impact assessments. To counter this, post-processing steps, encompassing downscaling and bias correction, are crucial when employing GCM forecasts (Tuel et al., 2021). The main objective here is to refine the global, coarse-grained GCM data into higher-resolution forecasts. These refined forecasts are far more pertinent for predicting seasonal drought events at a regional and local scale within the MedR."

**Lines 601-602: This section was mainly discussing projections, however here we are back to talking about forecasts. It is important to clarify this point.**

Thank you for your insightful comment. We have reconsidered this section and made the necessary revisions to make clear the difference between projections and predictions. This section includes now three subsections to discuss each time scale and type of analysis separately.

See sections:

Lines 648-672: 5.1. Multi-Model Ensemble.

Lines 673-701: 5.2. Coupled hydrological models.

Lines 702-726: 5.3. Long term drought projection under climate change.

**Line 606: It is not clear why the word 'necessary' is used. It would be beneficial to provide a clearer and more substantiated explanation to support this statement.**

Thank you for highlighting the ambiguity in this sentence. We agree that the term 'necessarily' might have added confusion. Our intention was to express that while these variables (soil moisture, streamflow, groundwater level, and PET) are indeed integral parts of the hydrological cycle, their representations in GCMs are often not as accurate as we would desire, due to the complexity of these processes and the spatial scale at which GCMs operate. We have revised this sentence to clarify this:

Lines 674-677: On the other hand, GCMs often struggle to accurately represent some complex elements of the hydrological cycle, such as soil moisture, streamflow, groundwater level, and PET. The inherent complexities of these variables and the broad spatial scale of GCMs make it challenging to fully capture their behavior. This gap can limit the effectiveness of GCMs in drought prediction and modelling (Balting et al., 2021).

**Line 615: It is recommended to add a reference to adequately support this statement.**

Thank you for your recommendation. We agree that adding a reference would support our statement. Here's the revised sentence with references included:

Lines 684-686: Crop growth models can also be coupled with hydrological models to make an accurate prediction of agricultural drought and its impact on crop yields (Narasimhan and Srinivasan, 2005; Abhishek et al., 2021).

**Line 633: It would be important to specify whether seasonal forecasting or decadal predictions are being discussed in this section. Clarifying the time frame will help readers better understand the context of the discussion." Regarding the comparison between dynamic and empirical forecasts, it**

**would be useful to mention whether there are studies or articles that have made such a comparison. It is recommended to read the papers by Ceglar et al. (2017) and Turco et al. (2017b).**

Thank you for your valuable comments. The section you referred to generally discusses the improvements in climate modeling, without differentiating between seasonal forecasting and decadal predictions. However, your suggestion to specify the time frame is well taken and we have divided this section to three subsections to discuss each time scale and type of analysis separately.

As for the comparison between dynamic and empirical forecasts, we concur that references to studies that have conducted such comparisons would enrich the discussion. The papers you suggested are indeed relevant and we have incorporated insights from these sources into our manuscript. (see lines 736-743).

**Line 642: This statement is not entirely accurate. While it is true that statistical models rely on historical data to make predictions, it does not necessarily mean that they cannot take climate changes into account.**

Thank you for bringing attention to this aspect of our discussion. We agree with your point that it was an oversimplification to state that statistical models are incapable of considering climate changes.

While it's accurate that these models are grounded in historical data, it is indeed possible to adjust or calibrate statistical models to account for shifting climate patterns, if these changes are reflected in the data they are trained on. However, such models might still face challenges when predicting unprecedented events that have no historical analogues.

We have reviewed this paragraph in response to the previous comment see lines (736-739).

**Lines 644-646: There is constant confusion between projections and forecasts in this section.**

Thank you for your valuable feedback. We agree with your remarque and have reviewed this section to ensure differentiating between these two terms.

Lines 736-743: "While statistical models, when appropriately fine-tuned, can effectively predict seasonal drought events, a significant limitation arises from the non-stationary relationship between the predictors and predictands used in the forecasting process (AghaKouchak et al., 2022). This can limit their ability to accurately predict unprecedented drought events, which fall beyond the scope of their historical training data (Hao et al., 2018). On the other hand, dynamical models are proficient at capturing the nonlinear interactions among the atmosphere, land, and ocean, enhancing their ability to detect the onset of droughts (Turco et al., 2017b; Ceglar et al., 2017). However, despite their advanced capabilities, their forecast proficiency is generally constrained to a few months of lead time (Turco et al., 2017b)."

**Line 662: There are no pure dynamic models, as they usually apply bias adjustment techniques.**

We agree with your point. We have removed the word "purely" from the sentence to eliminate any potential confusion.

**Line 668: There is a lack of references to support this statement. It is recommended to consult the works of Dutra et al. (2014); Mo and Lyon (2015); Yuan and Wood (2013), among others.**

Thank you for your recommendation. We have carefully reviewed these references and we agree that they provide valuable insights in hybrid statistical-dynamical drought forecasting. In the revised manuscript, we have incorporated citations from these works to strengthen the validity and credibility of our statements.

Lines 762-775: "On global scale, Yan and Wood (2013) analyzed the capability of seasonal forecasting of global drought onset and found that despite climate models increasing drought detection, a significant proportion of onset events are still missed. Their findings underscore the urgent need for implementing reliable, skillful probabilistic forecasting methods to better manage the inherent uncertainties and potentially improve drought predictability. Dutra et al. (2014) confirmed that the uncertainty in long lead time forecasts suggested that drought onset might fundamentally be a stochastic problem. Mo and Lyon (2015) also found that improvements in near-real-time global precipitation observations could yield the most substantial advances in global meteorological drought prediction in the near term. This reinforces the notion that the effectiveness of dynamical models is fundamentally associated to the quality of initial data and the inherent stochastic nature of drought onset."

**Lines 684-687: It is suggested to be consistent in the number of decimal places used, either 0 or 1 decimal place, throughout this segment of the text.**

Thank you for your comment. We agree with your suggestion. In the revised manuscript, we have ensured to standardize the decimal places across all values, and we have now limited them to two decimal places as suggested.

**Lines 686-696: Explanation of the indexes used, including SPI, and other related terms, should be made the first time they are mentioned in the text. This explanation should be in the discussion section.**

Thank you for your valuable comment. We agree that the explanation of the drought indices, including the SPI, would be more appropriately placed earlier in the manuscript. As per your suggestion, we have revised the manuscript to introduce and explain this index the first time it appears in the text, specifically within the section on meteorological drought.

Lines 147-153: "By fitting a probability distribution to observed precipitation data, the SPI is calculated and subsequently transformed into a standard normal distribution with a mean of 0 and a standard deviation of 1

(Livada and Assimakopoulos, 2007). Consequently, SPI values can be compared across various regions and timeframes (e.g., 1, 3, 6, 12, or 24 months). This multiscale nature of SPI enables it to capture diverse aspects of drought depending on the selected time scale. The shorter time scales (1-3 months) are suitable for monitoring agricultural drought, while longer time scales (6-12 months or more) are better suited for evaluating hydrological drought."

Following this revision, the discussion section now primarily focuses on a critical interpretation of the results. We believe these changes enhance the clarity and logical flow of the manuscript, allowing readers to better understand the methods and findings.

Lines 792-802: "The SPI was the primary indicator, used in 70.59% of meteorological drought studies. But it also served as an indicator for hydrological and agricultural droughts, with usage rates of around 25.00% and 8.33%, respectively. Despite the apparent versatility of the SPI, its reliance on precipitation data limits its ability to account for other influential factors such as evapotranspiration, soil moisture, land usage, and water management practices. Consequently, an overemphasis on the SPI could potentially constrain our comprehension of drought phenomena in the MedR. To enrich this understanding, it is recommended to incorporate a broader range of indicators and models that include a more diverse set of variables. Using multivariate drought indices such as the SPEI, PDSI, or sc-PDSI, or alternatively, a combination of multiple indices, can contribute to a more comprehensive view by including regional feedback mechanisms in the forecast process. This approach also enhances our capacity to evaluate the impacts of global warming on drought severity and intensity in the MedR (see Marcos-Garcia et al., 2017; Gouveia et al., 2017)."

**Lines 731-732: This sentence is not clear, since they do appear in Figure 4.**

Thank you for your feedback. We recognize the potential ambiguity around the term "hybrid models" due to its dual usage in the literature.

Typically, "hybrid models" may refer to statistical-dynamical models, combining statistical and dynamical approaches, as detailed in our paper in section 6. Alternatively, this term may denote the integration of two statistical models, an approach we have explored in section 4.3. In figure 4, the term "hybrid model" indeed pertains to the latter concept. We understand how our initial terminology could have suggested an integration of both statistical and dynamical methods, which was not our intent. To eliminate any potential confusion, we have updated the figure, its caption, and the associated text to specify these models as "hybrid statistical models.

**Line 735: It would be important to indicate the number of studies included in each boxplot to provide complete information on the analysis performed. In addition, it is suggested to include the ESP method in the analysis (for example, Torres-Vázquez et al., 2023). On the other hand, it would also be**

**interesting to distinguish between the different dynamic or hybrid methods used in the analysis, in addition to the statistical methods.**

Thank you for your insightful comments. We agree with your suggestion regarding the presentation of the number of studies included in each boxplot. In order to provide comprehensive details of our analysis, we have now attached a supplementary table in the appendix. This table explicitly lists all studies we compared, offering a more in-depth perspective on the body of research informing each boxplot.

Regarding the inclusion of the ESP method, you raise a valid point. We acknowledge that the ESP method (for example, as presented in Torres-Vázquez et al., 2023 and Turco et al., 2017b) could indeed be an interesting addition to our study. Therefore, we have supplemented our manuscript with a reference to the potential of the ESP method for seasonal drought forecasting, and provided a description of its principles, strengths, and limitations in section 4.5 as presented below.

Lines 70-73: "The climatology-based or persistence-based models, like the Ensemble Streamflow Prediction (ESP) system, form an essential tool in this category, leveraging both historical and near real-time data to generate a probabilistic forecast of future drought events (AghaKouchak, 2014a; Turco et al., 2017b; Torres-Vázquez et al., 2023)."

Line 569-595:

"**4.5. Ensemble Streamflow Prediction**

The ESP method, a commonly used technique in hydrological forecasting, was primarily intended for medium to long-term streamflow prediction (Day, 1985). However, its utility extends to the prediction of hydrological droughts, characterized by low streamflows (KyungHwan and DegHyo, 2015 ; Sutanto et al., 2020 ; Troin et al., 2021).

ESP operates on the principle of employing historical data to generate an ensemble of possible future climate conditions (Turco et al., 2017b). The process begins by determining the current state of the system, considering parameters such as current streamflow, soil moisture levels, and reservoir levels which serves as the initial conditions for the forecast (Wood et al., 2016). The generation of the ensemble involves choosing a historical record at each time (day, week or month) of forecast that will provide the meteorological inputs (Day, 1985). By repeating this process for every time in the historical record, an ensemble of forecasts is produced, each member representing a potential future scenario. The hydrological model is run for each ensemble member, using the chosen meteorological inputs and initial conditions to generate a range of potential future states of the system (Harrigan et al., 2018). The ensemble of forecasts is then analyzed to derive probabilistic predictions.

As new data becomes available, forecasts can be updated by re-initializing the system's state and generating a new ensemble of forecasts. A significant advantage of this method is that it enables the uncertainty prediction

by producing a variety of potential future streamflow forecast scenarios which can increase the confidence of this approach, specifically for its operational use in water management (Troin et al., 2021).

However, the limitations of the ESP method must be noted. For instance, it presupposes that future behavior will mirror past behavior, a concept that may not hold under changing climatic conditions (Wood et al., 2016). Furthermore, the method's performance is heavily reliant on the quality and duration of the historical meteorological records used in the ensemble generation process (Clark and Kavetski, 2010).

ESP is frequently employed as a benchmark for comparison with more sophisticated forecasting methods, such as dynamical climate models or hybrid statistical-dynamical models (AghaKouchak, 2014a; Turco et al., 2017b; Torres-Vázquez et al., 2023). Although these more complex methods can outperform ESP in some instances, the computationally efficient ESP method often exhibits comparable performance, particularly when forecasting a few months ahead (Turco et al., 2017b; Torres-Vázquez et al., 2023)."

Regarding the inclusion of ESP in figure 4 analysis, unfortunately, the geographical distribution of studies employing the ESP method for drought forecasting, as outlined by Troin et al. (2021), predominantly encompass North America (41%), Europe (28%), and Asia (21%). While the ESP method has demonstrated usefulness in various climatic contexts, its application to drought forecasting in the MedR has been minimal. As our work primarily aims to assess and compare the drought forecasting methods employed within the MedR, the scarcity of studies utilizing ESP in this specific context renders it incorporation in figure 4. This emphasizes the need for further research and application of the ESP method in Mediterranean climates to enhance the comprehensiveness of future comparative analyses.

Concerning your suggestion to distinguish between different dynamical or hybrid methods, we understand the value such a distinction would provide. However, due to the relatively limited number of studies using specific dynamic and hybrid approaches, a fine-grained breakdown might lead to categories with very small sample sizes, potentially skewing the results. In the revised manuscript, we have mentioned this limitation and expressed the need for more studies using diverse dynamical and hybrid models.

**Lines 737-740: It would be more appropriate to include this information in the legend of Figure 4.**

Thank you for your suggestion. We agree that placing this information within the legend of Figure 4 would enhance readability.

The caption reads now: lines 838-841: "Figure 4 Box and whiskers plot showing the performance of drought prediction models denoted by the coefficient of determination ($R^2$) for the surveyed studies in MedR. The lower box shows the 25th percentile, the upper box shows the 75 percentile and the median (50th percentile) is represented by the black line inside the box. The whiskers show the extent to the minimum and maximum values within 1.5 times the interquartile range (IQR) from the box."

**Line 774: Is the major interest only in hybrid models and not in dynamic models?**

Thank you for your comment. The remark in line 774 was intended to emphasize the emerging interest in hybrid models, considering their capacity to leverage the strengths of both statistical and dynamical models, potentially offering improved predictive accuracy. However, this does not diminish the importance or the ongoing research in dynamic models.

We have modified the sentence as follows:

Lines 874-877: "Notwithstanding these challenges, there is an increasing interest not only in enhancing traditional dynamical models but also in the development and utilization of hybrid models. As research progresses and resources become more accessible, these hybrid models may see wider adoption for their potential to improve predictive accuracy."

**Line 882: It is recommended to consult the works of Beck et al. (2019b) and of Turco et al. (2020).**

We appreciate your recommendation to consult the studies of Beck et al. (2019b) and Turco et al. (2020). We have found these references insightful and have integrated their key findings to support the statement in lines 974-977. This integration helps to underscore the importance of comprehensive and reliable data in improving the accuracy of drought monitoring and forecasting. Thank you for your constructive feedback.

Lines 984-987: "In addition, satellite observations of precipitation and soil moisture such as IMERG (Huffman et al., 2015), PERSIANN-CCS (Sadeghi et al., 2021), CHIRPS (Funk et al., 2015), SMAP (Entekhabi et al., 2010), MSWEP V2 (Beck et al., 2019), GLEAM v3 (Martens et al., 2017), and DROP (Turco et al., 2020) can be used in conjunction with the in-situ observations and ground-based radar observations data to fill observational gaps."

**Line 893: A detailed reading on GLEAM and SAFRAN is recommended. For instance, it is recommended Tramblay et al. (2019), Miralles et al. (2010) and Martens et al. (2017).**

Thank you for your insightful comment and the suggested literature to enhance our discussion on remote sensing data and reanalysis tools in drought forecasting and monitoring. We completely agree with your recommendation. We have since taken the time to review the works of Tramblay et al. (2019), Miralles et al. (2010), and Martens et al. (2017) for a deeper understanding of GLEAM and SAFRAN. These studies have provided invaluable insights into the utilization of these tools in bridging the gaps in observational data, aiding in the identification of potential drought risk areas, and tracking the progression of drought conditions over time.

Considering this, we have incorporated the knowledge garnered from these studies in the revised manuscript as follows:

Lines 984-987: "In addition, satellite observations of precipitation and soil moisture such as IMERG (Huffman et al., 2015), PERSIANN-CCS (Sadeghi et al., 2021), CHIRPS (Funk et al., 2015), SMAP (Entekhabi et al., 2010), MSWEP V2 (Beck et al., 2019), GLEAM v3 (Martens et al., 2017), and DROP (Turco et al., 2020) can be used in conjunction with the in-situ observations and ground-based radar observations data to fill observational gaps."

Lines 996-1002: "Similarly, SAFRAN, a high-resolution meteorological reanalysis, has shown its utility in regions with sparse observational data. Tramblay et al. (2019) used SAFRAN to generate a high-resolution (5 km) gridded daily precipitation dataset for Tunisia between 1979 and 2015. Their study, which combined data from 960 rain gauges with the SAFRAN analysis, demonstrated that SAFRAN surpassed other standard interpolation methods like Inverse Distance, Nearest Neighbors, Ordinary Kriging, or Residual Kriging with altitude. The outcome was a highly accurate gridded precipitation dataset that could be instrumental for climate studies, model evaluation, and hydrological modeling to support the planning and management of surface water resources."

**Lines 968-975: There is a clear confusion between projections and predictions.**

Thank you for pointing out this confusion. We have revised this paragraph to remove any ambiguity and ensure the appropriate use of these terms.

Lines 1080-1088: Dynamical models, given their ability to capture nonlinear interactions across the atmosphere, land, and ocean, offer considerable potential for more accurate and reliable seasonal drought predictions. However, the inherent chaotic nature of the atmosphere restricts their forecast skill to a few months in advance. The field has seen notable advancements, such as enhanced climate resolution, refined representation of physical processes, improved initialization methods, the application of multi-model ensembles, and the development of coupled modeling approaches. These developments have indeed bolstered the accuracy and reliability of drought predictions. Nevertheless, the implementation of these models in the MedR is constrained by challenges such as limited data availability, computational complexity, and inherent model uncertainties.

**Line 997: In reference to the cartographic rigor mentioned in this line, it would be useful to provide a more detailed explanation of what this refers to.**

Thank you for your comment. We agree with your suggestion regarding the need for a more detailed explanation of cartographic rigor in presenting probabilistic drought risk maps and we have now expanded this statement.

The term "cartographic rigor" in this context refers to the accurate representation of spatial data in a manner that is both precise and informative. This includes correct geospatial positioning, appropriate scale and projection, and the intuitive use of symbols, colors, or contours with a clearly defined legend to effectively illustrate the range of probabilities. Probabilistic information, representing the uncertainty of predictions, should

be conveyed in an easily interpretable manner, such as through color gradations, to reflect different probability levels.

The updated text reads as follows:

Lines 1109-1112: "An informative visualization of results via probabilistic drought risk maps is recommended, whereby color gradations or contouring are used to effectively illustrate the range of probabilities. Ensuring cartographic rigor, such maps should maintain spatial accuracy, use appropriate scaling, and include a clearly defined legend to decrypt different probability levels."

**References**

[revised manuscript text omitted]

WMO, G., 1994. Guide to hydrological practices: data aquisition and processing, analysis, forecasting and other applications.

Wood, A.W., Hopson, T., Newman, A., Brekke, L., Arnold, J., Clark, M., 2016. Quantifying Streamflow Forecast Skill Elasticity to Initial Condition and Climate Prediction Skill. Journal of Hydrometeorology 17, 651–668. https://doi.org/10.1175/JHM-D-14-0213.1